environmental science/meteorology/energy

fixed-tilt solar panel, monofacial photovoltaics, solar energy, clouds

**Author for correspondence:**
Gábor Horváth
e-mail: gh@arago.elte.hu

# How the morning-afternoon cloudiness asymmetry affects the energy-maximizing azimuth direction of fixed-tilt monofacial solar panels

Péter Takács[1], Judit Slíz-Balogh[1,2], Ákos Horváth[3], Dániel Horváth[1], Imre M. Jánosi[4,5] and Gábor Horváth[1]

[1]Environmental Optics Laboratory, Department of Biological Physics, and [2]Department of Astronomy, ELTE Eötvös Loránd University, H-1117 Budapest, Pázmány sétány 1, Hungary
[3]Meteorological Institute, Universität Hamburg, Bundesstrasse 55, D-20146 Hamburg, Germany
[4]Max Plack Institute für Physik Komplexer Systeme, Nöthnitzer Strasse 38, D-01187 Dresden, Germany
[5]University of Public Service, Faculty of Water Science, Ludovika tér 2, H-1083 Budapest, Hungary

PT, 0000-0002-5668-0901; JS-B, 0000-0001-8883-1973;
ÁH, 0000-0002-5860-2368; IMJ, 0000-0002-3705-5748;
GH, 0000-0002-9008-2411

In the Northern Hemisphere, south is the conventional azimuth direction of fixed-tilt monofacial solar panels, because this orientation may maximize the received light energy. How does the morning-afternoon cloudiness asymmetry affect the energy-maximizing azimuth direction of such solar panels? Prompted by this question, we calculated the total light energy received by a fixed-tilt monofacial solar panel in a whole year, using the celestial motion of the Sun and the direct and diffuse radiation measured hourly throughout the year in three North American (Boone County, Tennessee, Georgia) and European (Italy, Hungary, Sweden) regions. Here we show that, depending on the tilt angle and the local cloudiness conditions, the energy-maximizing ideal azimuth of a solar panel more or less turns eastward from south, if afternoons are cloudier than mornings in a yearly average. In certain cases, the turn of the ideal azimuth of such solar panels may be worth taking into consideration, even though the maximum energy gain is not larger than 5% for nearly vertical panels. Specifically, when solar panels are fixed on vertical walls or oblique roofs with non-ideal tilt, the deviation of the energy-maximizing azimuth from the south can be incorporated in the design of buildings.

# 1. Introduction

In the Northern/Southern Hemisphere, fixed-tilt monofacial solar panels conventionally face south/ north, because the southern/northern azimuth may ensure maximal solar energy [1–9]. Monofacial panels collect light only from their photovoltaic front side, while bifacial panels use special solar cells and a transparent cover to collect light not only from the front, but also from the rear side [10].

All else being equal, how does the morning-afternoon cloudiness asymmetry affect the energy-maximizing azimuth of such solar panels? To answer this question, we determined the total light energy $e$ available for a fixed-tilt monofacial solar panel throughout the year. Our calculations were performed as functions of the elevation angle $\theta$ (=$90° - \beta$, where $\beta$ is the panel's tilt angle measured from the horizontal) and the azimuth angle $\alpha$ of the panel's normal vector, separately for three Northeast American (Boone County, Tennessee and Georgia) and three European (Italy, Hungary, Sweden) regions. Based on the solar movement in the sky and radiometric data (direct sunlight and diffuse skylight) for these six different geographical areas, we show that due to the morning-afternoon cloudiness asymmetry the energy-maximizing azimuth of fixed-tilt monofacial solar panels in the Northern Hemisphere deviates from south by an amount that depends on the tilt angle. Here we demonstrate that there may be some practical advantage to varying the orientation of photovoltaic solar panels from direct south/north if there is systematic morning/afternoon asymmetry in cloudiness.

# 2. Material and methods

## 2.1. Calculation of the solar elevation and azimuth angles versus time

Using the method of Bretagnon and Francou [11], the time-dependent elevation angle $\theta_s(t)$ and azimuth angle $\alpha_s(t)$ of the Sun on the celestial hemisphere were calculated as described by Horváth *et al.* [12].

## 2.2. Calculation of the light energy absorbed by a Fresnel-reflecting ($R > 0$) fixed-tilt monofacial solar panel

We consider the sky radiation absorbed by a fixed-tilt monofacial solar panel only from sunrise to sunset because the light energy absorbed between sunset and sunrise is negligible relative to the daylight absorption [13]. According to figure 1, the unit normal vector of the fixed-tilt solar panel is:

$$\boldsymbol{n} = (\cos\theta_n \cdot \sin\alpha_n, \cos\theta_n \cdot \cos\alpha_n, \sin\theta_n), \tag{2.1}$$

where axes $x$ and $y$ point to west and south, axis $z$ points vertically upward to the zenith, the elevation angle $\theta_n$ ($\geq 0°$) of $\boldsymbol{n}$ is measured from the horizontal, and the azimuth angle $\alpha_n$ is measured clockwise from axis $y$ pointing south. For fixed-tilt solar panels $\theta_n$ and $\alpha_n$ are constant. The unit vector pointing toward the Sun is (figure 1):

$$\boldsymbol{s} = (\cos\theta_s \cdot \sin\alpha_s, \cos\theta_s \cdot \cos\alpha_s, \sin\theta_s), \tag{2.2}$$

where $\theta_s$ ($\geq 0°$) is the solar elevation angle from the horizontal, and $\alpha_s$ is the azimuth angle measured clockwise from south. The global irradiance received from the celestial hemisphere by a horizontal surface is $I_{global} = I_{Sun} + I_{diff}$, where $I_{Sun}$ and $I_{diff}$ are the direct (sunlight) and diffuse (skylight) irradiances measured in Joule/ second/$m^2$/nanometer by a horizontal sensor surface. The total light energy absorbed by a fixed-tilt solar panel between dawn and dusk on the i-th day (counted from 1 January) is the sum of the energy $E_{Sun,i}(\theta_n, \alpha_n)$ absorbed from direct sunlight and the energy $E_{diff,i}(\theta_n, \alpha_n)$ absorbed from diffuse skylight:

$$E_i = E_{Sun,i}(\theta_n, \alpha_n) + E_{diff,i}(\theta_n, \alpha_n). \tag{2.3}$$

In figure 2 let us consider point P of the sky-dome on a circle, the plane of which tilts with angle $\eta$ from the horizontal and the position vector of P has an angle $\beta$ from the position vector of point B being in the vertical $x$–$z$ plane on the circle. The unit vector $\boldsymbol{p}$ pointing to P is (figure 2):

$$\boldsymbol{p} = (\cos\beta \cdot \cos\eta, \sin\beta, \cos\beta \cdot \sin\eta). \tag{2.4}$$

In figure 2 the unit normal vector $\boldsymbol{m}$ of the fixed-tilt solar panel is:

$$\boldsymbol{m} = (\cos\theta_n, 0, \sin\theta_n). \tag{2.5}$$

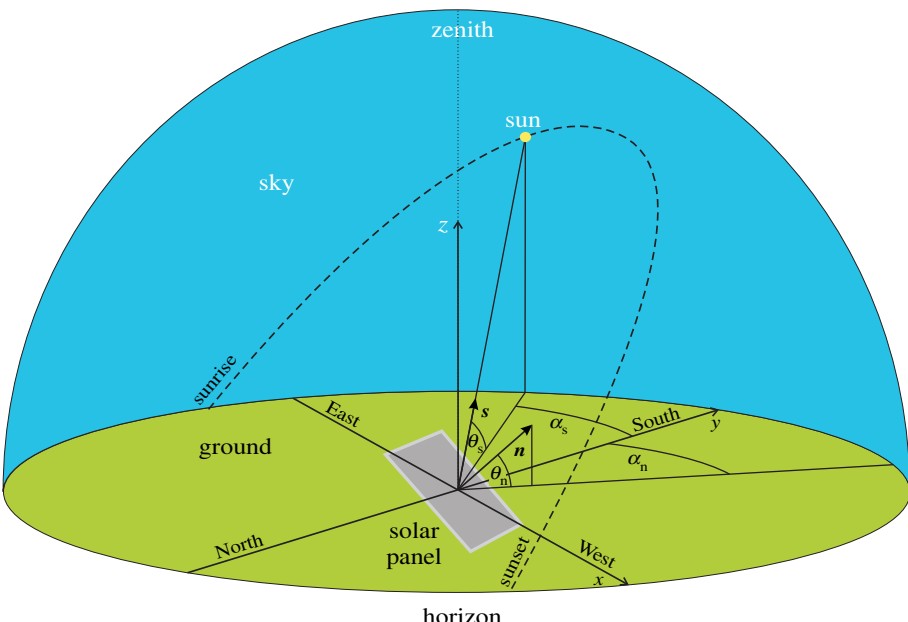

**Figure 1.** Geometry of a fixed-tilt monofacial solar panel receiving sunlight and skylight. The unit normal vector of the panel's surface is **n**, and the unit vector **s** points toward the Sun.

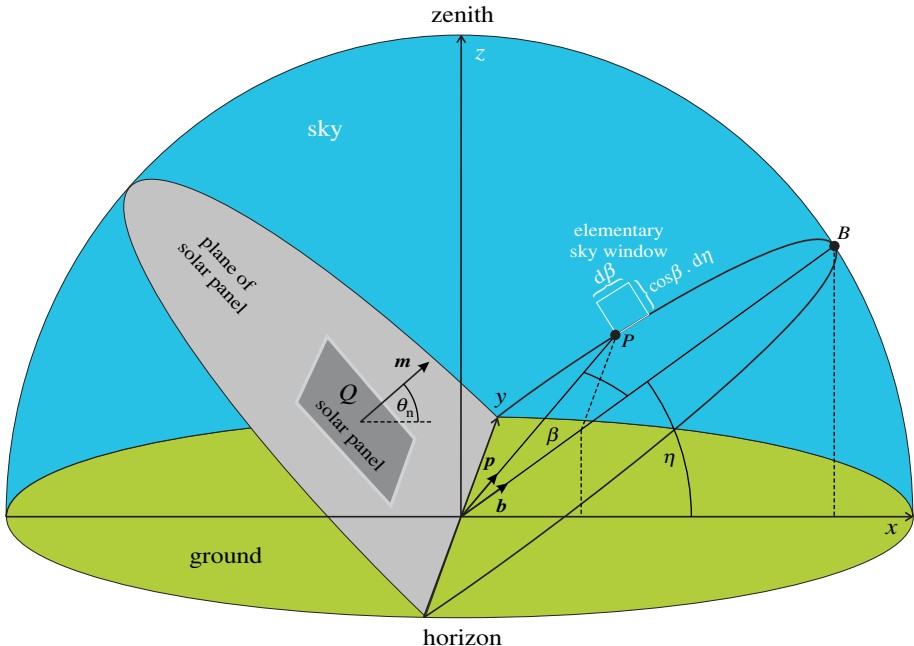

**Figure 2.** Geometry of the celestial hemisphere with a fixed-tilt monofacial solar panel, where **m** is the panel's unit normal vector and $Q$ is its surface area. Point P of the sky-dome is on a circle, the plane of which tilts with angle $\eta$ from the horizontal, and the unit vector **p** pointing to P has an angle $\beta$ from the unit vector **b** pointing to point B being in the vertical $x$–$z$ plane on the circle.

The infinitesimal energy $\delta E_{\mathrm{diff}}$ received by surface $Q$ of the solar panel from the diffuse skylight with irradiance $I_{\mathrm{diff}}(\lambda)$ within an infinitesimal time period $\mathrm{d}t$, in the infinitesimal wavelength interval $\mathrm{d}\lambda$ through the elementary sky window with angular dimension $\mathrm{d}\beta \cdot \cos\beta \cdot \mathrm{d}\eta$ at P is:

$$\delta E_{\mathrm{diff}} = I_{\mathrm{diff}}(\lambda) \cdot Q \cdot \mathrm{d}t \cdot \mathrm{d}\lambda \frac{\cos\beta \cdot \mathrm{d}\beta \cdot \mathrm{d}\eta}{2\pi}, \tag{2.6}$$

where $2\pi$ is the angular extension of the celestial hemisphere (figure 2). The diffuse skylight incident on the solar panel splits into two components (figure 3$a$):

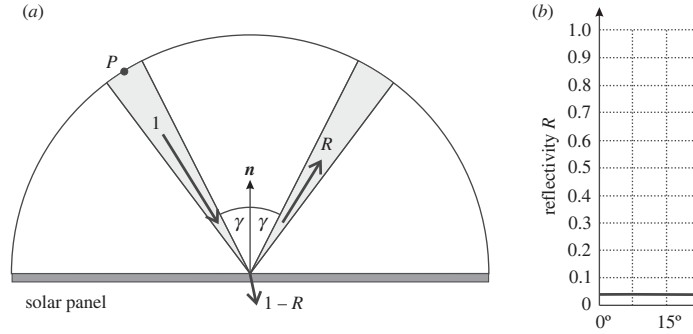

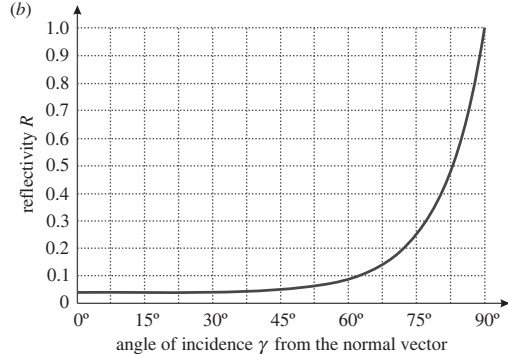

**Figure 3.** (*a*) A unit amount 1 of diffuse skylight incident on a solar panel splits into two components: proportion $R$ is Fresnel-reflected from the weather-proof smooth dielectric layer, and proportion $1 - R$ penetrates into the absorber, where $R$ is the Fresnel's reflectivity. (*b*) Reflectivity $R(\gamma)$ of a glass/plastic dielectric surface with $n_d = 1.5$ and $n_a = 1$, where $\gamma$ is the incidence angle from the normal vector of the surface [14, pp. 50–76].

(i) **The first component** is Fresnel-reflected from the weather-proof smooth outermost dielectric layer, the Fresnel's reflectivity $R$ of which is [14]:

$$
\left.
\begin{aligned}
R(\cos\gamma) &= \frac{[\rho_{\text{para}}(\cos\gamma)]^2 + [\rho_{\text{perp}}(\cos\gamma)]^2}{2}, \\[2mm]
\rho_{\text{para}}(\cos\gamma) &= \frac{n_a\cos\gamma - \sqrt{n_d^2 - n_a^2(1-\cos^2\gamma)}}{n_a\cos\gamma + \sqrt{n_d^2 - n_a^2(1-\cos^2\gamma)}}, \\[2mm]
\rho_{\text{perp}}(\cos\gamma) &= \frac{n_d^2\cos\gamma - n_a\sqrt{n_d^2 - n_a^2(1-\cos^2\gamma)}}{n_d^2\cos\gamma + n_a\sqrt{n_d^2 - n_a^2(1-\cos^2\gamma)}},
\end{aligned}
\right\}
\tag{2.7}
$$

and

where $\rho_{\text{para}}$ and $\rho_{\text{perp}}$ are the amplitude reflection coefficients for parallel and perpendicular polarization (of incident light) with respect to the surface, $n_a$ and $n_d$ are the refractive indices of air and dielectric, and $\gamma$ is the angle of incidence measured from the normal vector $\boldsymbol{n}$ of the reflecting surface. The direct sunlight is unpolarized, while scattered diffuse skylight is more or less linearly polarized, depending on the celestial direction from which it originates. It would be exceedingly difficult to account for the polarization pattern of the sky depending on many factors, especially on the Sun's position, cloudiness, wavelength and aerosol concentration [15]. Thus, using (2.7) in our model calculation, we assumed that the diffuse light received by the solar panel is unpolarized. We took $n_a = 1$ (air) and $n_d = 1.5$ (glass/plastic), while the dispersion (wavelength dependence) of dielectric can be neglected in a first approximation.

(ii) **The second component** is transmitted by the dielectric layer (figure 3*a*). The smooth, Fresnel-reflecting dielectric with reflectivity $R(\cos\gamma)$ transmits $1 - R(\cos\gamma)$ proportion of the incident light towards the underlying absorber layer, the absorption spectrum of which is $0 \leq A(\lambda, \gamma) \leq 1$, where $\gamma$ is the incidence angle from the normal vector of the surface. Thus, the net absorbance of the solar panel is:

$$
A_{\text{net}} = [1 - R(\cos\gamma)] \cdot A(\lambda, \gamma).
\tag{2.8}
$$

Later on (see subsection 2.4) we consider only the case $A(\lambda, \gamma) = 1$, because we calculate the maximal possible total light energy per unit area available for a Fresnel-reflecting ($R > 0$) fixed-tilt solar panel integrated for the whole year. Figure 3*b* displays the reflectivity curve $R(\gamma)$ of a glass/plastic dielectric surface with $n_d = 1.5$ in air with $n_a = 1$ calculated from (2.7). Using (2.6) and (2.8), the elementary energy absorbed from diffuse skylight by the surface $Q$ of the solar panel is:

$$
dE_{\text{diff}} = \frac{I_{\text{diff}}(\lambda) \cdot Q \cdot dt \cdot d\lambda \cdot \cos\beta \cdot d\beta \cdot d\eta \cdot [1 - R(\cos\gamma_{\text{mp}})] \cdot A(\lambda, \gamma)}{2\pi},
\tag{2.9}
$$

where $\gamma_{\text{mp}}$ is the incidence angle between unit vectors $\boldsymbol{m}$ and $\boldsymbol{p}$. Using (2.4) and (2.5), the cosine of $\gamma_{\text{mp}}$ is:

$$
\cos\gamma_{\text{mp}}(\theta_n, \beta, \eta) = \boldsymbol{m} \cdot \boldsymbol{p} = \cos\beta \cdot (\cos\theta_n \cdot \cos\eta + \sin\theta_n \cdot \sin\eta).
\tag{2.10}
$$

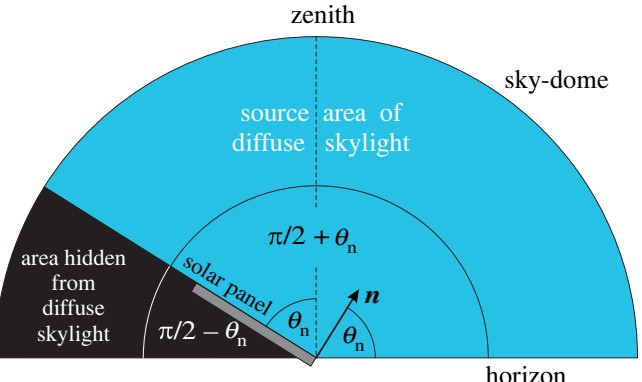

**Figure 4.** For calculation of the proportion $(\theta_n + \pi/2)/\pi$ of the celestial hemisphere from which a fixed-tilt monofacial solar panel with elevation angle $\theta_n$ of its normal vector **n** receives diffuse skylight (blue). From the black area of the sky-dome the panel does not receive diffuse skylight.

To obtain the energy component absorbed by the solar panel from diffuse skylight on the i-th day, $dE_{\text{diff}}$ expressed by (2.9) should be integrated (i) temporally from time $t_{\text{rise}}^i$ of sunrise to time $t_{\text{set}}^i$ of sunset, (ii) spatially for the sky region $-\pi/2 \leq \beta \leq +\pi/2$ and $0 \leq \eta \leq \theta_n + \pi/2$ from which the panel receives diffuse skylight (figures 2 and 4) and (iii) spectrally for the wavelength range $\lambda_{\min} \leq \lambda \leq \lambda_{\max}$ in which the panel's absorbance is relevant:

$$E_{\text{diff,i}}(\theta_n, \alpha_n) = Q\tau_{\text{diff}}(\theta_n) \int_{t_{\text{rise}}^i}^{t_{\text{set}}^i} \left\langle \int_{\lambda_{\min}}^{\lambda_{\max}} A(\lambda, \gamma) \cdot I_{\text{diff}}(\lambda, t) \cdot d\lambda \right\rangle dt, \tag{2.11}$$

with

$$\tau_{\text{diff}}(\theta_n) = \frac{1}{2\pi} \int_{\eta=0}^{\eta=\theta_n+\pi/2} \left\{ \int_{\beta=-\pi/2}^{\beta=+\pi/2} [1 - R(\theta_n, \beta, \eta)] \cdot \cos\beta \cdot d\beta \right\} d\eta$$

$$= \frac{2\theta_n + \pi}{2\pi} - \frac{1}{2\pi} \int_{\eta=0}^{\eta=\theta_n+\pi/2} \left\{ \int_{\beta=-\pi/2}^{\beta=+\pi/2} R(\theta_n, \beta, \eta) \cdot \cos\beta \cdot d\beta \right\} d\eta, \tag{2.12}$$

where $\tau_{\text{diff}}(\theta_n)$ is the net transmissivity of the panel's dielectric layer for diffuse skylight, $I_{\text{diff}}(\lambda, t)$ is the diffuse irradiance received by a horizontal surface, and $\lambda_{\min} = 200$ nm $\leq \lambda \leq \lambda_{\max} = 4000$ nm is the solar-energetically relevant wavelength interval of sky radiation [1].

From (2.11) we obtain the diffuse light energy per unit area absorbed by the solar panel:

$$e_{\text{diff,i}}(\theta_n, \alpha_n) = \frac{E_{\text{diff,i}}(\theta_n, \alpha_n)}{Q} = \tau_{\text{diff}}(\theta_n) \int_{t_{\text{rise}}^i}^{t_{\text{set}}^i} \left\langle \int_{\lambda_{\min}}^{\lambda_{\max}} A(\lambda, \gamma) \cdot I_{diff}(\lambda, t) \cdot d\lambda \right\rangle dt. \tag{2.13}$$

The elementary direct sunlight energy $dE_{\text{Sun}}$ absorbed by the solar panel in an infinitesimal time interval $dt$ and in an infinitesimal wavelength range $d\lambda$ is:

$$dE_{\text{Sun}} = Q \cdot \cos\gamma_{\text{ns}} \cdot I_{\text{Sun}}(\lambda, \theta_s) \cdot A(\lambda, \gamma) \cdot d\lambda \cdot dt, \tag{2.14}$$

where $\gamma_{\text{ns}}$ is the incidence angle between unit vectors **n** and **s**. Using (2.1) and (2.2), the cosine of $\gamma_{\text{ns}}$ is:

$$\cos\gamma_{\text{ns}} = \boldsymbol{n} \cdot \boldsymbol{s} = \cos\theta_n \cdot \sin\alpha_n \cdot \cos\theta_s \cdot \sin\alpha_s + \cos\theta_n \cdot \cos\alpha_n \cdot \cos\theta_s \cdot \cos\alpha_s + \sin\theta_n \cdot \sin\theta_s. \tag{2.15}$$

The solar panel can absorb direct sunlight only if the following condition is satisfied:
$-90° < \gamma_{\text{ns}} < +90°$, $0 < \cos\gamma_{\text{ns}} < 1$, that is

$$0 < \cos\theta_n \cdot \sin\alpha_n \cdot \cos\theta_s \cdot \sin\alpha_s + \cos\theta_n \cdot \cos\alpha_n \cdot \cos\theta_s \cdot \alpha_s + \sin\theta_n \cdot \sin\theta_s < 1. \tag{2.16}$$

Using (2.8) and (2.14), the direct solar energy per unit area absorbed by the solar panel is:

$$e_{\text{sun,i}}(\theta_n, \alpha_n) = \frac{E_{\text{sun,i}}(\theta_n, \alpha_n)}{Q} = \int_{t_{\text{rise}}^i}^{t_{\text{set}}^i} \left\langle \cos\gamma_{\text{ns}}(t)\{1 - R[\cos\gamma_{\text{ns}}(t)]\} \int_{\lambda_{\min}}^{\lambda_{\max}} A(\lambda, \gamma) I_{\text{Sun}}[\lambda, \theta_s^i(t)] \, d\lambda \right\rangle dt, \tag{2.17}$$

where factor

$$\cos\gamma_{ns}(t) = \cos\theta_n \sin\alpha_n \cos\theta_s^i(t) \sin\alpha_s^i(t) + \cos\theta_n \cos\alpha_n \cos\theta_s^i(t) \cos\alpha_s^i(t) + \sin\theta_n \sin\theta_s^i(t), \quad (2.18)$$

is necessary, because the direct solar radiation is usually not perpendicular to the panel's surface. Finally, the total light energy $e$ per unit area absorbed by the solar panel from 1 January to 31 December is:

$$e = \sum_{i=1}^{i=365} e_i = \sum_{i=1}^{i=365} [e_{\text{Sun},i}(\theta_n, \alpha_n) + e_{\text{diff},i}(\theta_n, \alpha_n)], \quad (2.19)$$

where $i = 1$ and $i = 365$ denote the first (1 January) and the last (31 December) day of the year, and components $e_{\text{diff},i}(\theta_n, \alpha_n)$ and $e_{\text{Sun},i}(\theta_n, \alpha_n)$ are expressed by (2.13) and (2.17), respectively.

## 2.3. ERA5 radiation data

Direct and diffuse insolation data are from the ERA5 (European Re-Analysis, generation 5) of the European Centre for Medium-Range Weather Forecasts (ECMWF) [16]. It combines model data with global observations into a complete and consistent gridded ($0.25° \times 0.25°$) dataset on 137 pressure levels between the surface and 1 Pa, from 1950 onward with a temporal resolution of 1 hour. The native time variable assigned to all data is in UTC (Universal Time Coordinated).

Two products are used in the present study for the wavelength range $\lambda_{\min} = 200$ nm $\leq \lambda \leq \lambda_{\max} = 4000$ nm: (i) shortwave solar radiation downward (SSRD), representing the amount of downward flux of solar radiation on a horizontal unit surface. This parameter comprises both direct and diffuse solar radiation. Radiation from the Sun (solar or shortwave radiation) is partly reflected back to space by clouds and aerosol particles in the atmosphere and some of it is absorbed. The rest is incident on the Earth's surface (represented by this parameter). To a reasonably good approximation, this parameter is the model equivalent of what would be measured by a thermopile pyranometer. (ii) The second product is the ERA5 direct solar radiation at the surface (FDIR), the amount of direct radiation reaching a horizontal unit surface area. Note that both parameters are accumulated over the hour and thus given in units of J m$^{-2}$. To convert to W m$^{-2}$, the accumulated values need to be divided by 3600 s.

To a reasonably good approximation, the difference SSRD–FDIR is what would be measured by a diffuse pyranometer. But care must be taken that the direct (i.e. non-scattered) solar radiation in the model actually includes radiation that has been scattered by cloud particles by a fraction of a degree, since the scattering pattern of cloud particles has a narrow peak in the forward direction. A further point is that the diffuse downwelling radiation from the model includes diffuse radiation in the direction of the Sun, which would be excluded from the radiation measured by a diffuse pyranometer that uses a shadow band to exclude direct radiation. However, since we make estimates for solar panels where no area is shadowed out in the direction of the Sun, corrections are not implemented.

In the ERA5 radiation scheme, incoming solar radiation is attenuated by absorbing gases (water vapour, carbon dioxide, methane, ozone, other trace gases) and is scattered by molecules, aerosols and cloud particles [17]. For water vapour and clouds, the radiation scheme uses prognostic information from the forecast model. For ozone, only diagnostic values are used (i.e. ozone has no feedback on the atmosphere via the radiation scheme); however, ozone profiles, total column ozone estimates and ozone-sensitive channel radiances from a large number of sub-daily satellite observations are assimilated in the reanalysis. The spatial and seasonal distribution of greenhouse gases ($CO_2$, $CH_4$, $N_2O$, CFC-11, CFC-12) are prescribed by monthly zonally averaged concentration profiles. The blocking of solar radiation by aerosols is described by climatological distributions of optical depth from sea salt, soil/dust, black carbon and sulfate (including stratospheric sulfate from major volcanic eruptions of the last century). Input are monthly mean geographical profiles of optical depth, which account for large-scale seasonal variations. The contribution of local diurnal variations in aerosol optical depth, which is the only major radiative effect missing from ERA5, is discussed in §4.

As for the reliability of ERA5 radiation data, there are some (mostly local) validations and intercomparisons with other reanalyses. One of the most comprehensive recent reviews by Yang & Bright [18] compared six new generation satellite-derived datasets and two reanalyses, ERA5 and MERRA-2 (Modern-Era Retrospective analysis for Research and Applications, v. 2) with 27 years of continuous terrestrial observations on 57 reference sites, with hourly resolution. Satellite data are difficult to compare with reanalyses (they provide neither spatial nor temporal global coverage), but the final conclusion of [18] is that ERA5 clearly outperforms MERRA-2. More restricted regional

comparisons have very similar conclusions, e.g. over the Indonesian region [19]. Two recent validations using Chinese records observed larger errors; however, they noted that cloudy-rainy regions showed the largest deviations, which is the consequence of the relatively poor representation of clouds in all global weather forecast and climate models [20,21]. Overall, ERA5 currently represents the most accurate global description of the state of the atmosphere.

## 2.4. Maximum light energy available for Fresnel-reflecting ($R > 0$) fixed-tilt monofacial solar panels

From the ERA5 radiation data, we determined the mean power flux $W_{\mathrm{Sun}}$ (W m$^{-2}$) of direct sunlight and the mean power flux $W_{\mathrm{diff}}$ (W m$^{-2}$) of diffuse skylight averaged for the period 2009–2019 and measured by a horizontal detector surface for the 24 one-hour ($\Delta t = 1$ h) intervals of the day ($1 \leq k \leq 24$) in Boone County (39.0° N, −84.75° E, local time: UTC − 5 h), Tennessee (35.5° N, −88.25° E, UTC − 6 h), Georgia (31.25° N, −83.25° E, UTC − 5 h), Central Italy (41.0° N, 15.0° E, UTC + 1 h), Central Hungary (47.0° N, 19.0° E, UTC + 1 h) and South Sweden (58.0° N, 13.0° E, UTC + 1 h) (electronic supplementary material, figures S1–S14, tables S1–S12). With the use of these power fluxes, the maximal possible—when $A(\lambda, \gamma) = 1$—total light energy per unit area available for a Fresnel-reflecting ($R > 0$) fixed-tilt monofacial solar panel integrated for the whole year is:

$$e(\theta_{\mathrm{n}}, \alpha_{\mathrm{n}}, R > 0, A = 1) = \sum_{i=1}^{i=365} [e_{\mathrm{Sun},i}(\theta_{\mathrm{n}}, \alpha_{\mathrm{n}}, R > 0, A = 1) + e_{\mathrm{diff},i}(\theta_{\mathrm{n}}, \alpha_{\mathrm{n}}, R > 0, A = 1)], \quad (2.20)$$

where the direct sunlight energy component is:

$$e_{\mathrm{Sun},i}(\theta_{\mathrm{n}}, \alpha_{\mathrm{n}}, R > 0, A = 1) = \int_{t_{\mathrm{rise}}^{i}}^{t_{\mathrm{set}}^{i}} \{1 - R[\cos \gamma_{\mathrm{ns}}(t, \theta_{\mathrm{n}}, \alpha_{\mathrm{n}})]\} \cdot \cos \gamma_{\mathrm{ns}}(t, \theta_{\mathrm{n}}, \alpha_{\mathrm{n}}) \frac{W_{\mathrm{Sun}}[\theta_{\mathrm{s}}^{i}(t)]}{\sin \theta_{\mathrm{s}}^{i}(t)} \, dt, \quad (2.21)$$

where $W_{\mathrm{Sun}}$ has to be divided by $\sin\theta_{\mathrm{s}}$, because it is measured by a horizontal detector surface, while here we need the direct power flux perpendicular to the sunlight. In (2.21) the factor $\cos\gamma_{\mathrm{ns}}(t, \theta_{\mathrm{n}}, \alpha_{\mathrm{n}})$ is given by (2.18). For the above-mentioned three Northeast-American and three European regions we calculated the mean direct energy flux $W_{\mathrm{Sun}}(k) \cdot \Delta t$ for the $k$-th ($1 \leq k \leq 24$) one-hour period $\Delta t = 1$ h of the day, and transformed (2.21) to the following sum:

$$e_{\mathrm{Sun},i}(\theta_{\mathrm{n}}, \alpha_{\mathrm{n}}, R > 0, A = 1) = \sum_{k=1}^{k=24} \{1 - R[\cos \gamma_{\mathrm{ns}}(k, \theta_{\mathrm{n}}, \alpha_{\mathrm{n}})]\} \cdot \cos \gamma_{\mathrm{ns}}(k, \theta_{\mathrm{n}}, \alpha_{\mathrm{n}}) \frac{W_{\mathrm{Sun}}(k) \cdot \Delta t}{\sin \theta_{\mathrm{s}}^{i}(k)},$$

$$\cos \gamma_{\mathrm{ns}}(k) = \cos \theta_{\mathrm{n}} \sin \alpha_{\mathrm{n}} \cos \theta_{\mathrm{s}}^{i}(k) \sin \alpha_{\mathrm{s}}^{i}(k) + \cos \theta_{\mathrm{n}} \cos \alpha_{\mathrm{n}} \cos \theta_{\mathrm{s}}^{i}(k) \cos \alpha_{\mathrm{s}}^{i}(k) + \sin \theta_{\mathrm{n}} \sin \theta_{\mathrm{s}}^{i}(k). \quad (2.22)$$

In (2.20) the diffuse skylight energy component is:

$$e_{\mathrm{diff},i}(\theta_{\mathrm{n}}, \alpha_{\mathrm{n}}, R > 0, A = 1) = \tau_{\mathrm{diff}}(\theta_{\mathrm{n}}) \int_{t_{\mathrm{rise}}^{i}}^{t_{\mathrm{set}}^{i}} W_{\mathrm{diff}}(t) \, dt, \quad (2.23)$$

where $\tau_{\mathrm{diff}}(\theta_{\mathrm{n}})$ is given by (2.12). For the three American and three European regions we calculated the mean diffuse energy flux $W_{\mathrm{diff}}(k) \cdot \Delta t$, where $\Delta t = 1$ h and $1 \leq k \leq 24$, and transformed (2.23) to the following sum:

$$e_{\mathrm{diff},i}(\theta_{\mathrm{n}}, \alpha_{\mathrm{n}}, R > 0, A = 1) = \tau_{\mathrm{diff}}(\theta_{\mathrm{n}}) \sum_{k=1}^{k=24} W_{\mathrm{diff}}(k) \cdot \Delta t. \quad (2.24)$$

## 2.5. Maximum light energy available for anti-reflective ($R = 0$) fixed-tilt monofacial solar panels

Until now we have dealt with fixed-tilt monofacial solar panels having a smooth, Fresnel-reflecting ($R > 0$) cover surface. Nowadays, solar panels with an anti-reflective (matte) cover surface are gradually spreading [22–25]. The reflectivity of such matte covers is very small: $0 < R \ll 1$. Using $R(\lambda) = 0$ and $A(\lambda) = 1$ in (2.22) and (2.24), we obtain the possible maximum total light energy $e_{\mathrm{matte}}$ per

unit area available for an ideal matte solar panel integrated for the whole year:

$$e_{\text{matte}} = \sum_{i=1}^{i=365} [e_{\text{Sun,matte},i}(\theta_{\text{n}}, \alpha_{\text{n}}, R = 0, A = 1) + e_{\text{diff,matte},i}(\theta_{\text{n}}, \alpha_{\text{n}}, R = 0, A = 1)], \tag{2.25}$$

where the components $e_{\text{Sun,matte},i}(\theta_{\text{n}}, \alpha_{\text{n}}, R = 0, A = 1)$ and $e_{\text{diff,matte},i}(\theta_{\text{n}}, \alpha_{\text{n}}, R = 0, A = 1)$ are expressed as follows:

$$e_{\text{Sun,matte,i}}(\theta_{\text{n}}, \alpha_{\text{n}}, R = 0, A = 1) = \sum_{k=1}^{k=24} \cos \gamma_{\text{ns}}(k, \theta_{\text{n}}, \alpha_{\text{n}}) \frac{W_{\text{Sun}}(k) \cdot \Delta t}{\sin\theta_{\text{s}}^{i}(k)}, \tag{2.26}$$

and

$$e_{\text{diff,matte},i}(\theta_{\text{n}}, \alpha_{\text{n}}, R = 0, A = 1) = \frac{2\theta_{\text{n}} + \pi}{2\pi} \sum_{k=1}^{k=24} W_{\text{diff}}(k) \cdot \Delta t. \tag{2.27}$$

# 3. Results

Figure 5 shows the total light energy $e$ per unit area available for a fixed-tilt monofacial solar panel with Fresnel's reflectivity $R(\lambda)$ greater than 0 (figure 3b) between 1 January and 31 December in Boone County (39.0° N, −84.75° E), Tennessee (35.5° N, −88.25° E), Georgia (31.25° N, −83.25° E), Central Italy (41.0° N, 15.0° E), Central Hungary (47.0° N, 19.0° E) and South Sweden (58.0° N, 13.0° E) as functions of the elevation angle $\theta_{\text{n}}$ and the azimuth angle $\alpha_{\text{n}}$ of the panel's normal vector. Apart from South Sweden, the distribution of the $e$ values is asymmetric to the geographical south ($\alpha_{\text{n}} = 0°$) due to the yearly average asymmetric daily morning-afternoon cloudiness. Since in Boone County, Tennessee, Georgia, Central Italy and Central Hungary afternoons are in yearly average cloudier than mornings, the energetically ideal $\theta*_{\text{n}}(\alpha*_{\text{n}})$ curve—along which $e$ is maximal for a given $\theta*_{\text{n}}$—runs in the eastern (left) half of figure 5a–e between $\alpha*_{\text{n}}(\theta*_{\text{n}} = 85°) = -3°/-2°/-5°/0°/-4°$ and $\alpha*_{\text{n}}(\theta*_{\text{n}} = 0°) = -31°/-36°/-42°/-25°/-21°$ (electronic supplementary material, tables S13, S14). However, since in South Sweden mornings are in yearly average as cloudy as afternoons, the ideal azimuth angle is practically south $\alpha*_{\text{n}} \approx 0°$, independently of the elevation $\theta*_{\text{n}}$ (figure 5f) (electronic supplementary material, table S14).

The above computations were repeated for a fixed-tilt solar panel with zero reflectivity $R(\lambda) = 0$. According to figure 6, the results are qualitatively the same as for a Fresnel-reflecting fixed-tilt solar panel (figure 5), but the numerical values are slightly different (electronic supplementary material, tables S15, S16): the $\theta*_{\text{n}}(\alpha*_{\text{n}})$ curve for Boone County, Tennessee, Georgia, Central Italy and Central Hungary runs again in the eastern (left) half of figure 6a–e between $\alpha*_{\text{n}}(\theta*_{\text{n}} = 85°) = -3°/-2°/-5°/0°/-4°$ and $\alpha*_{\text{n}}(\theta*_{\text{n}} = 0°) = -22°/-27°/-34°/-15°/-15°$. In South Sweden $\alpha*_{\text{n}} \approx 0°$ again for any $\theta*_{\text{n}}$ (figure 6f, electronic supplementary material, table S16).

Figure 7a displays the percent energy gain $\Delta Z_{\text{ms}}(\theta*_{\text{n}}) = (e_{\text{max}} - e_{\text{south}})/e_{\text{south}}$ of a Fresnel-reflecting fixed-tilt solar panel with reflectivity $R(\lambda)$ greater than 0 (figure 3b) and ideal elevation angle $\theta*_{\text{n}}$ and azimuth angle $\alpha*_{\text{n}}$ of the panel's normal vector compared to a panel with the same $\theta*_{\text{n}}$ but facing south ($\alpha*_{\text{n}} = 0°$) in Boone County, Tennessee, Georgia, Central Italy, Central Hungary and South Sweden, where $e_{\text{max}} = e(\theta*_{\text{n}}, \alpha*_{\text{n}})$ and $e_{\text{south}} = e(\theta*_{\text{n}}, \alpha_{\text{n}} = 0°)$. $\Delta Z_{\text{ms}}$ decreases systematically from 5% (Georgia), 2.8% (Tennessee), 1.8% (Boone County), 0.8% (Central Hungary) and 0.6% (Central Italy) to zero as $\theta*_{\text{n}}$ increases from 0° to 90°, while in South Sweden it has a negligible maximum of 0.04% at $\theta*_{\text{n}} = 20°$ (figure 7a; electronic supplementary material, tables S13, S14).

Finally, we studied an anti-reflective solar panel with zero reflectivity $R(\lambda) = 0$. According to figure 7b, the results are similar to those for a Fresnel-reflecting panel (figure 7a): $\Delta Z_{\text{ms}}$ decreases systematically from 2.7% (Georgia), 1.3% (Tennessee), 0.9% (Boone County), 0.5% (Central Hungary) and 0.1% (Central Italy) to zero as $\theta*_{\text{n}}$ increases from 0° to 90°, while in South Sweden it has a maximum of 0.09% at $\theta*_{\text{n}} = 10°$ (figure 7b, electronic supplementary material, tables S15, S16).

From the above results, our conclusions are the following:

— If mornings are less cloudy than afternoons averaged for the whole year in a region, then the energy-maximizing ideal azimuth angle $\alpha*_{\text{n}}$ of fixed-tilt solar panels deviates from south ($\alpha_{\text{n}} = 0°$) toward east by an angle varying between 1° and 42°, depending on the elevation angle $\theta*_{\text{n}}$ of the panel's normal vector ranging between 0° and 90°.

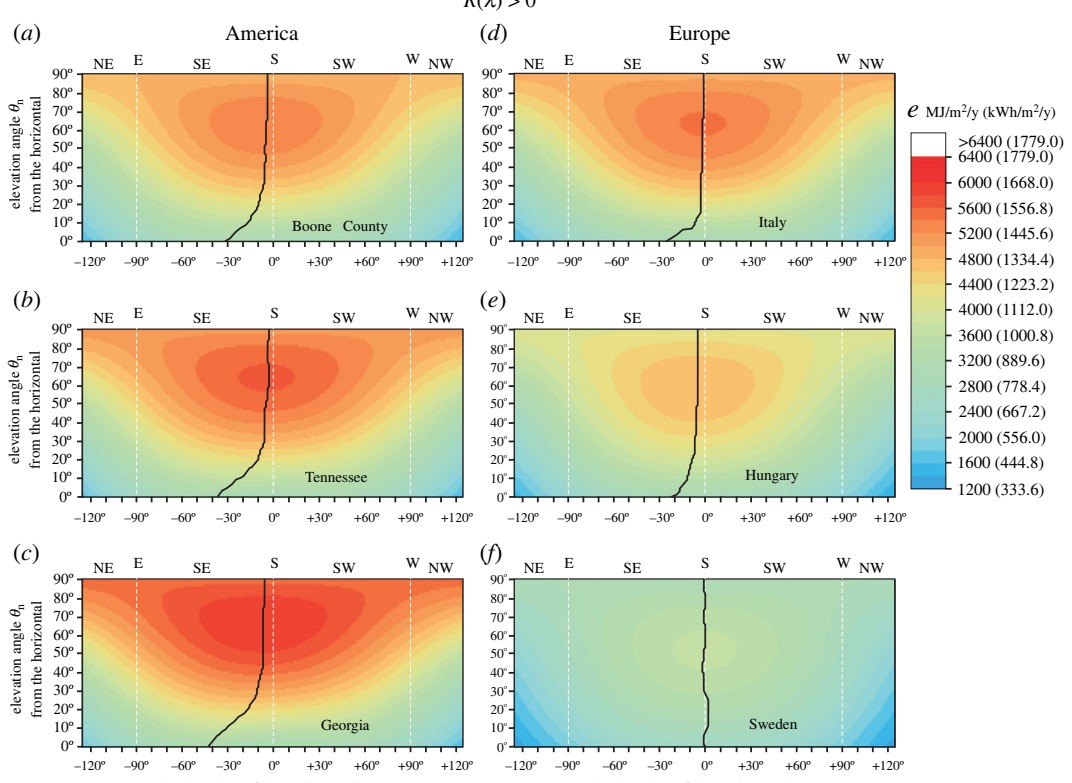

**Figure 5.** Colour-coded values of the total light energy $e$ (in MJ/m$^2$/year as well as kWh/m$^2$/year) per unit area available for a Fresnel-reflecting (reflectivity $R > 0$) fixed-tilt monofacial solar panel between 1 January and 31 December in (a) Boone County (39.0° N, −84.75° E), (b) Tennessee (35.5° N, −88.25° E), (c) Georgia (31.25° N, −83.25° E), (d) Central Italy (41.0° N, 15.0° E), (e) Central Hungary (47.0° N, 19.0° E) and (f) South Sweden (58.0° N, 13.0° E), as functions of the elevation angle $\theta_n$ (from the horizontal) and the azimuth angle $\alpha_n$ (clockwise from the geographical south) of the panel's normal vector. The Fresnel's reflectivity $R(\lambda) > 0$ of the smooth outer surface of the solar panel is shown in figure 3b. The black continuous curves mark the ideal $(\theta*_n, \alpha*_n)$ angle pairs for which $e$ is maximal for a given $\theta*_n$.

— If mornings are as cloudy as afternoons in yearly average, then the energy-maximizing ideal azimuth angle $\alpha*_n$ of fixed-tilt solar panels is south ($\alpha_n = 0°$), independently of the panel's elevation angle $\theta*_n$.
— Depending on cloudiness (i.e. geographical region) and $\theta*_n$ of a fixed-tilt solar panel with energy-maximizing ideal azimuth angle $\alpha*_n$, the energy gain $\Delta Z_{ms}$ is not larger than about 5% compared to a panel with the same elevation $\theta*_n$ but facing south ($\alpha_n = 0°$).

## 4. Discussion

In this work, we showed that the azimuth orientation of an energy-maximizing (ideal) fixed-tilt monofacial solar panel deviates by 1°–42° (depending on the tilt angle) from the geographical south toward east, if in yearly average mornings are less cloudy than afternoons. If in yearly average mornings and afternoons are equally cloudy, the ideal azimuth is south, while if mornings are cloudier than afternoons, the ideal azimuth turns westward. The smaller the elevation angle $\theta_n$ of the panel's normal vector from the horizontal, the larger the deviation of the ideal azimuth from south.

We investigated two models of fixed-tilt monofacial solar panels: (1) the Fresnel-reflecting smooth outer surface of the panel had a reflectivity $R(\gamma)$ greater than 0, where $\gamma$ is the incidence angle (figure 3b), and (2) the anti-reflective (matte) outer surface had zero reflectivity $R(\gamma) = 0$. Considering the dependence of the total light energy absorbed by these panels throughout the year on the elevation angle $\theta_n$ and the azimuth angle $\alpha_n$ of the panel's normal vector, both panel types are qualitatively very similar (figures 5–7). The only important quantitative difference between them is that type 2 absorbs more light energy than type 1.

Fixed-tilt solar panels are designed such that the elevation $\theta_n \geq 45°$ of their normal vector ensures a maximum energy-producing efficiency. In this range of $\theta_n$, depending on the local cloud conditions, the maximum deviation of the ideal azimuth from south toward east is not larger than 8° with a very small

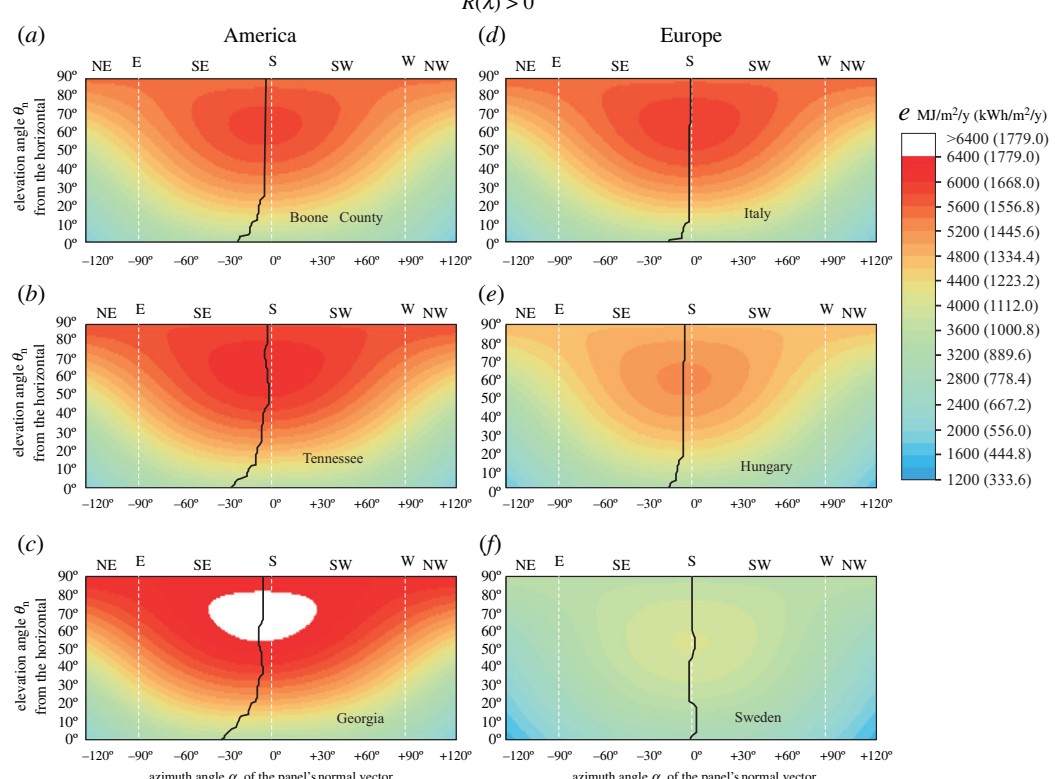

**Figure 6.** The same as figure 5 but for an anti-reflective solar panel with zero reflectivity $R(\lambda) = 0$.

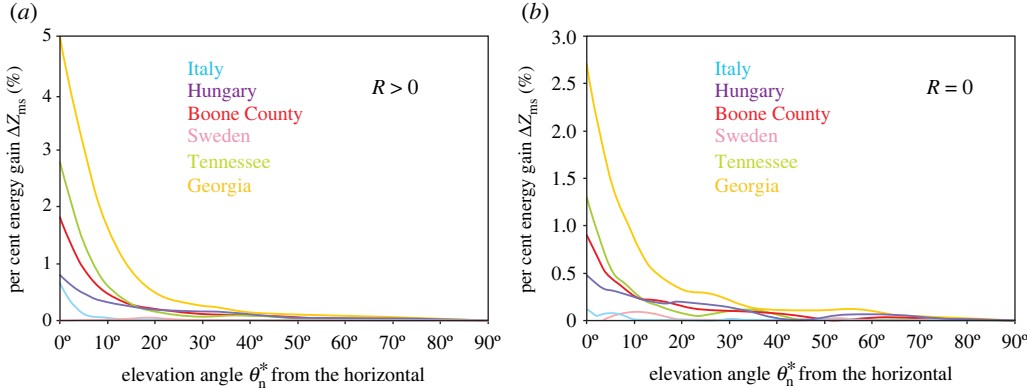

**Figure 7.** Percent energy gain $\Delta Z_{ms}(\theta*_n) = (e_{max} - e_{south})/e_{south}$ of a fixed-tilt monofacial solar panel with ideal elevation angle $\theta*_n$ (from the horizontal) and azimuth angle $\alpha*_n$ (clockwise from the geographical south) of the panel's normal vector compared to a panel with the same $\theta*_n$ but facing south ($° \alpha*_n = 0°$) in Boone County (39.0° N, −84.75° E), Tennessee (35.5° N, −88.25° E), Georgia (31.25° N, −83.25° E), Central Italy (41.0° N, 15.0° E), Central Hungary (47.0° N, 19.0° E) and South Sweden (58.0° N, 13.0° E), where $e_{max} = e(\theta*_n, \alpha*_n)$ and $°e_{south} = e(\theta*_n, \alpha_n = 0°)$. (a) Fresnel-reflecting panel with reflectivity $R(\lambda) > 0$ shown in figure 3b. (b) Anti-reflective panel with zero reflectivity $R(\lambda) = 0$.

energy gain of $\Delta Z \leq 0.1\%$ compared to a solar panel facing south with the same tilt (electronic supplementary material, tables S13–S16). Thus, in such cases the energy gain is practically negligible.

However, let us consider fixed-tilt solar panels with non-ideal tilts. Such panels are typically installed on oblique roofs or vertical walls. On roofs the elevation $\theta_n$ of the panel's normal vector can range between 0° and 90°, while on vertical walls $\theta_n = 0°$. In these cases, depending on the local cloud conditions, the maximum deviation of the ideal azimuth $\alpha*_n$ from south toward east is 15°–42° with a maximum energy gain of $\Delta Z = 5\%$ compared to a vertical panel facing south (electronic supplementary material, tables S13–S16). These angular deviations from south are already considerable and the corresponding energy gains may be worth utilizing.

The azimuth direction of roofs/walls of existing buildings cannot be changed. In this case, it is worth installing solar panels on those roofs/walls, whose azimuth is closest to the ideal azimuth $\alpha^*_n$, which can considerably deviate from south. New buildings, however, can purposefully be designed so that their solar panels face the ideal azimuth, rather than the conventional geographical south.

The current analysis focused on standard fixed-tilt monofacial solar panels, which are the dominant technology today. Bifacial solar panels are, however, getting increasing attention due to the potentially lower cost of electricity they offer for many locations in the world [10]. The computational methodology used in this work can also be applied to bifacial panels. Such a detailed analysis, which should take into consideration the absorption of earthlight (i.e. sky- and sunlight reflected from the ground) by the panel's rear side, is the topic of future research. Here we only mention the qualitative expectation that depending on the tilt angle, the energy-maximizing ideal azimuth of fixed-tilt bifacial panels also turns from the conventional geographical south in regions where afternoons are usually cloudier than mornings. The rear side of fixed-tilt bifacial panels with front side facing approximately south or southeast receives practically only diffuse skylight and earthlight, the irradiance of which is an order of magnitude less than that of direct sunlight. Thus, the ideal azimuth of such panels is also predominantly determined by sunlight and, therefore, similar to the azimuth of fixed-tilt monofacial panels.

In order to broaden the perspective of our analysis, it is worth performing similar computations for single-axis monofacial and bifacial tracking panels, which usually track the direct sunlight around the east-west axis [26]. For morning/evening cloudiness asymmetry scenarios, an intelligent tracking could be implemented to maximize the absorbed energy.

Let us consider the potential effects on our results of the two main limitations of the current study. First, although the ERA5 radiation calculations do account for the large-scale (geographical) and low-frequency (monthly) variability of aerosols, they neglect the local diurnal variation of aerosol loading, which mainly affects the direct solar component. Over most urban/industrial sites, the aerosol optical depth increases by 10–40% during the day with a maximum in the afternoon, as revealed by ground-based measurements from the Aerosol Robotic Network (AERONET) [27]. Thus, the irradiance of the dominant direct sunlight is slightly lower and the irradiance of the diffuse (aerosol-scattered) skylight is slightly higher in the afternoon than in the morning. This aerosol-induced asymmetry in morning-afternoon illumination is analogous to the asymmetry caused by the diurnal cycle of cloudiness and turns the ideal azimuth further east at most locations. At a few sites, however, local meteorology (e.g. afternoon sea breeze) can result in a decreasing aerosol loading during the day. An improved model could incorporate AERONET measurements to quantify the added eastward azimuth turn under typical conditions but also to account for atypical aerosol loads.

Second, the warming up of solar panels is known to degrade electric output, because conversion efficiency drops with temperature [28–31]. Vaillon et al. [29] listed three options to mitigate thermal effects in photovoltaic electric energy conversion. The first is to maximize cooling, the second is to minimize the thermal load in the panel, and the third is to minimize the thermal sensitivity of the electrical power output. In our current calculations, the temperature dependence of a solar cell's power generation efficiency is neglected. This conversion efficiency decreases/increases by 0.2–0.5% for every 1°C increase/decrease in temperature above/below the 25°C reference temperature used in standard test conditions [32]. The operational cell temperature is higher than the ambient air temperature during daytime and primarily depends on the thermal properties of the cell material, the geometry and orientation of the panel, the type of the background surface (roof, wall or open field), the solar insolation, and the amount of ventilation, which in turn depends on wind speed. Therefore, the actual temperature variation is highly location- and installation-specific. Nevertheless, both weather data-based thermal modelling studies and long-duration outdoor tests indicate that the diurnal cycle of cell temperature is usually skewed towards the afternoon, even in cloudy conditions: that is, the cell temperature is generally higher in the afternoon than in the morning [33–35]. The resulting (opposite) asymmetry in conversion efficiency, similar to the asymmetry in cloudiness and aerosol load, favours the morning, that is, the eastern hemisphere. Although a thermal loss of around 0.1–0.5%/K does not seem to be dramatic, it nevertheless needs to be investigated whether or not the 'optimal' orientation of solar panels—narrowly defined in the current study as the azimuth that maximizes the available solar energy—actually has a net positive effect on electric output. Such an empirical study is deferred to future research.

Taken together, the published observational data on the typical diurnal cycle of aerosol load and solar panel operating temperature as well as our ERA5-based radiation calculations strongly suggest the eastward turn of the ideal, energy-maximizing azimuth from due south, at locations where mornings

are less cloudy than afternoons. The ideal azimuth direction can be further refined, and the expected eastward turn confirmed if site-specific weather data are available at higher spatio-temporal resolutions than the ones provided by the global atmospheric reanalysis used in the current work.

Cloud cover reduces the annual insolation, and thus PV solar yield in general. However, clouds influence the efficiency of solar panels in other ways, too. In deserts, apart from the high temperatures (decreasing the efficiency of PV panels), dust and sand accumulation on the panels (decreasing the light intensity available for panels) is also of great concern. Depending on the frequency of windy conditions, the dust/sand-covered panels should periodically be cleaned, which is a time-consuming and expensive activity. On the other hand, countries with cloudy climates (e.g. Ireland, England, Scandinavia) usually experience stronger winds, cooler air temperatures, and more frequent rain (which cleans the solar panels); these factors increase PV electric output all else being equal.

Cloud transitions also affect the performance of PV systems. The irradiance incident on PV generators can considerably exceed the expected clear sky irradiance, a phenomenon called cloud enhancement (CE) [36]. Due to CE, the maximum power of the PV generator can exceed the rated power of the inverter connecting the generator to the grid. It was shown that the effect of CE is small on the aggregated energy because CE events that most strongly impact PV system operations are very rare [36].

The fast irradiance transitions caused by clouds are partial shading events that cause fast power fluctuations leading even to stability and quality problems in power networks [37]. Fast non-homogeneous irradiance transitions also cause mismatch losses in PV generators and the occurrence of multiple maximum power points (MPPs), which appear in a wide voltage range of the PV generator. It was demonstrated that the energy losses due to operation at a local MPP instead of the global one during partial shading events by clouds have only a minor effect on the total energy production of PV arrays, especially for large-scale systems [38].

In an improved model of our computational approach, the above effects can also be taken into consideration to determine the performance-maximizing (rather than the insolation-maximizing) locally ideal azimuth angle of solar panels.

Ethics. For our studies no permission, licence or approval was necessary.

Data accessibility. Our paper has the following electronic supporting materials: electronic supplementary material, tables S1–S16, figures S1–S14 [39].

Authors' contributions. P.T.: Conceptualization, data curation, formal analysis, investigation, methodology, software, validation, visualization, writing—original draft, writing—review and editing; J.S.: Conceptualization, data curation, formal analysis, investigation, methodology, project administration, software, validation, visualization, writing—original draft, writing—review and editing; A.H.: Data curation, investigation, resources, software, validation, writing—original draft, writing—review and editing; D.H.: Data curation, investigation, methodology, software, visualization; I.M.J.: Conceptualization, data curation, formal analysis, investigation, methodology, software, validation, writing—original draft, writing—review and editing; G.H.: Conceptualization, data curation, formal analysis, funding acquisition, investigation, methodology, resources, supervision, validation, visualization, writing—original draft, writing—review and editing.

All authors gave final approval for publication and agreed to be held accountable for the work performed therein.

Conflict of interest declaration. The authors have no competing interests.

Funding. There was no funding.

Acknowledgements. We are grateful to Prof. Bálint Érdi (Department of Astronomy, Eötvös University, Budapest) for his algorithm of the Sun's orbit relative to the Earth that was used in our computer model. We thank two anonymous referees and the subject editor, Peter Haynes, for their constructive reviews.

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
