## [Peer Review File · Royal Society Open Science]

Review History

RSOS-210406.R0 (Original submission)

Review form: Reviewer 1

Is the manuscript scientifically sound in its present form?

Yes

Are the interpretations and conclusions justified by the results?

Yes

Is the language acceptable?

Yes

Do you have any ethical concerns with this paper?

No

Have you any concerns about statistical analyses in this paper?

No

Recommendation?

Accept with minor revision (please list in comments)

Comments to the Author(s)

The paper explores the effect of morning-afternoon cloudiness asymmetry on the energy-maximizing azimuth direction of solar panels. The authors calculate the deviation of the energy maximizing azimuth from the standard south orientation which is typically used in the northern hemisphere. The paper is well-written, and the methodologies and results are discussed in sufficient detail. The topic addressed in the paper is novel and important from both practical and intellectual perspectives related to the understanding and deployment of solar photovoltaics systems. I have a few questions/comments:

1) The analysis has been done on standard (monofacial) panels which are the dominant technology for today. Bifacial solar panels are however getting much attraction due to the reducing costs and potentially lower levelized cost of electricity for many locations of the world. See for example: Rodríguez-Gallegos, Carlos D., et al. "Global techno-economic performance of bifacial and tracking photovoltaic systems." *Joule* 4.7 (2020): 1514-1541.

While it may require substantial additions to extend the computations done by the authors from the monofacial to bifacial, the authors may want to make a comment whether the same methodology could be applied for bifacial panels or not. Is there any qualitative insight they could provide for South faced fixed tilt bifacial panels with relatively simple extrapolation? A special case of vertically mounted East/West facing bifacial panels may be discussed relatively easily within the mathematical framework of the paper since this case may be approximated by a superposition of East facing monofacial (during mornings) and west faced monofacial (during afternoons).

2) Related to the previous comment, a similar comment may be relevant for the single axis tracking panels which usually track the direct beam around the East/West axis. For morning/evening asymmetry scenario, an intelligent tracking may be implemented to maximize energy (see for example: Patel, M. Tahir, et al. "Global analysis of next-generation utility-scale PV: Tracking bifacial solar farms." *Applied Energy* 290 (2021): 116478.). Authors may want to refer this paper to broaden the readers' perspective.

3) The authors state that the temperature and dust effects could be assumed symmetric for morning-afternoons. I wonder whether the cloudiness asymmetry for morning-evening has any effect on temperature? A common perception is that the cloudy periods during the daytime are relatively lower in temperature compared to the otherwise sunny conditions. Can the authors verify the annual symmetry for temperature and aerosol particulate matter through the typical meteorological data for the locations explored in the paper?

Review form: Reviewer 2**Is the manuscript scientifically sound in its present form?**

Yes

Are the interpretations and conclusions justified by the results?

Yes

Is the language acceptable?

Yes

Do you have any ethical concerns with this paper?

No

Have you any concerns about statistical analyses in this paper?

No

Recommendation?

Major revision is needed (please make suggestions in comments)

Comments to the Author(s)

1. "Fixed-tilt solar panels conventionally face south." That's not true in Australia, where they're facing north (and towards the Equator). Please correct.
2. "Mature sunflower inflorescences absorb maximal light energy, if they face east" Yes, the mature sunflower is not heliotropic. Yet, sunflowers do track the sun during the day from east to west at the bud stage, i.e. when they are in need of great energy resources. Please incorporate this important detail in your analysis.
3. Reference [1] is quoted, but Ref [1] uses a range from $\lambda_{\min}=170$ nm to $\lambda_{\max}=10$ μm , contradicting the here selected "relevant wavelength interval of sky radiation". Actually, $\lambda_{\min} = 250$ nm is often not relevant, since light with $\lambda < 380$ nm is typically absorbed by the glass / encapsulant layers. In addition, for silicon solar cells $\lambda_{\max} = 1200$ nm $\gg 900$ nm. In conclusion, the authors seem to have not only opted for a too narrow bandwidth interval by including the often irrelevant UV range (250 nm .. 380 nm). How will the results change for a more appropriate spectral range, e.g. from 400 nm to 1200 nm?
4. Why does the absorption spectrum A in Eq 8 not depend on the angle of incidence (AOI)? The factor $1-R$ accounts for the external reflection losses, but the escape reflection losses do greatly depend on the absorber thickness and the AOI. Please clarify.
5. Since $A(\lambda)$ is here independent of the AOI, the study solely focuses on the external reflection losses and thus underestimates the total reflection, i.e. it overestimates the total absorption. Despite this point becomes irrelevant, because the authors use $A=1$ later on, it should be made clear in the text that only external reflection losses were considered in the calculations.
6. Please convert MJ/cm² into kWh/year/cm².
- 7a. As solar cells are encapsulated in PV modules, their operating temperature is in general higher than the ambient temperature. This is especially the case in the afternoon, when more heat is radiated out by Earth's surface, once the local insolation has passed its peak value. However, higher temperatures can lead to significant increases in the dark-current and in turn to a reduction in the power conversion efficiency. How do you separate the impact of morning-afternoon ambient temperature from the morning-afternoon cloudiness?
- 7b. You state that the effect of temperature does not affect your main conclusions, because "in yearly average the neglected [temperature] effects influence the morning and afternoon photovoltaic efficiencies equally." Without numbers, this statement is a speculation and does not justify the neglect of temperature dynamics. More important of whether two effects influence the PV efficiencies equally or unequally is to quantify/estimate the magnitude of these two effects.

Hence, what influences the azimuth angle the most? Is it the morning-afternoon ambient temperature, or is it the morning-afternoon cloudiness?

Decision letter (RSOS-210406.R0)

Dear Dr Horvath

The Editors assigned to your paper RSOS-210406 "How the morning-afternoon cloudiness asymmetry affects the energy-maximizing azimuth direction of fixed-tilt solar panels" have now received comments from reviewers and would like you to revise the paper in accordance with the reviewer comments and any comments from the Editors. Please note this decision does not guarantee eventual acceptance.

Please submit your revised manuscript and required files (see below) no later than 21 days from today's (ie 12-Oct-2021) date. Note: the ScholarOne system will 'lock' if submission of the revision is attempted 21 or more days after the deadline. If you do not think you will be able to meet this deadline please contact the editorial office immediately.

on behalf of Peter Haynes (Subject Editor)
openscience@royalsociety.org

Associate Editor Comments to Author:

Comments to the Author:

There are a number of matters that you will need to address before the paper can be considered ready for publication; however, the changes needed do not - on the face of it - appear to be onerous. Given the concerns raised, we'd like you to revise the paper (supplying both a tracked-changes version of the revision and a thorough point-by-point response) before you resubmit. Good luck!

Reviewer comments to Author:

Reviewer: 1

Comments to the Author(s)

The paper explores the effect of morning-afternoon cloudiness asymmetry on the energy-maximizing azimuth direction of solar panels. The authors calculate the deviation of the energy maximizing azimuth from the standard south orientation which is typically used in the northern hemisphere. The paper is well-written, and the methodologies and results are discussed in sufficient detail. The topic addressed in the paper is novel and important from both practical and intellectual perspectives related to the understanding and deployment of solar photovoltaics systems. I have a few questions/comments:

1) The analysis has been done on standard (monofacial) panels which are the dominant technology for today. Bifacial solar panels are however getting much attraction due to the reducing costs and potentially lower levelized cost of electricity for many locations of the world. See for example: Rodríguez-Gallegos, Carlos D., et al. "Global techno-economic performance of bifacial and tracking photovoltaic systems." *Joule* 4.7 (2020): 1514-1541.

While it may require substantial additions to extend the computations done by the authors from the monofacial to bifacial, the authors may want to make a comment whether the same methodology could be applied for bifacial panels or not. Is there any qualitative insight they could provide for South faced fixed tilt bifacial panels with relatively simple extrapolation? A special case of vertically mounted East/West facing bifacial panels may be discussed relatively easily within the mathematical framework of the paper since this case may be approximated by a superposition of East facing monofacial (during mornings) and west faced monofacial (during afternoons).

2) Related to the previous comment, a similar comment may be relevant for the single axis tracking panels which usually track the direct beam around the East/West axis. For morning/evening asymmetry scenario, an intelligent tracking may be implemented to maximize energy (see for example: Patel, M. Tahir, et al. "Global analysis of next-generation utility-scale PV: Tracking bifacial solar farms." *Applied Energy* 290 (2021): 116478.). Authors may want to refer this paper to broaden the readers' perspective.

3) The authors state that the temperature and dust effects could be assumed symmetric for morning-afternoons. I wonder whether the cloudiness asymmetry for morning-evening has any effect on temperature? A common perception is that the cloudy periods during the daytime are relatively lower in temperature compared to the otherwise sunny conditions. Can the authors verify the annual symmetry for temperature and aerosol particulate matter through the typical meteorological data for the locations explored in the paper?

Reviewer: 2

Comments to the Author(s)

1. "Fixed-tilt solar panels conventionally face south." That's not true in Australia, where they're facing north (and towards the Equator). Please correct.

2. "Mature sunflower inflorescences absorb maximal light energy, if they face east" Yes, the mature sunflower is not heliotropic. Yet, sunflowers do track the sun during the day from east to west at the bud stage, i.e. when they are in need of great energy resources. Please incorporate this important detail in your analysis.

3. Reference [1] is quoted, but Ref [1] uses a range from $\lambda_{\min}=170$ nm to $\lambda_{\max}=10$ μm , contradicting the here selected "relevant wavelength interval of sky radiation". Actually, $\lambda_{\min} = 250$ nm is often not relevant, since light with $\lambda < 380$ nm is typically absorbed by the glass / encapsulant layers. In addition, for silicon solar cells $\lambda_{\max} = 1200$ nm $\gg 900$ nm. In conclusion, the authors seem to have not only opted for a too narrow bandwidth interval by including the often irrelevant UV range (250 nm .. 380 nm). How will the results change for a more appropriate spectral range, e.g. from 400 nm to 1200 nm?

4. Why does the absorption spectrum A in Eq 8 not depend on the angle of incidence (AOI)? The factor $1-R$ accounts for the external reflection losses, but the escape reflection losses do greatly depend on the absorber thickness and the AOI. Please clarify.

5. Since $A(\lambda)$ is here independent of the AOI, the study solely focuses on the external reflection losses and thus underestimates the total reflection, i.e. it overestimates the total absorption. Despite this point becomes irrelevant, because the authors use $A=1$ later on, it should be made clear in the text that only external reflection losses were considered in the calculations.

6. Please convert MJ/cm² into kWh/year/cm².

7a. As solar cells are encapsulated in PV modules, their operating temperature is in general higher than the ambient temperature. This is especially the case in the afternoon, when more heat is radiated out by Earth's surface, once the local insolation has passed its peak value. However, higher temperatures can lead to significant increases in the dark-current and in turn to a reduction in the power conversion efficiency. How do you separate the impact of morning-afternoon ambient temperature from the morning-afternoon cloudiness?

7b. You state that the effect of temperature does not affect your main conclusions, because "in yearly average the neglected [temperature] effects influence the morning and afternoon photovoltaic efficiencies equally." Without numbers, this statement is a speculation and does not justify the neglect of temperature dynamics. More important of whether two effects influence the PV efficiencies equally or unequally is to quantify/estimate the magnitude of these two effects. Hence, what influences the azimuth angle the most? Is it the morning-afternoon ambient temperature, or is it the morning-afternoon cloudiness?

===PREPARING YOUR MANUSCRIPT===

===PREPARING YOUR REVISION IN SCHOLARONE===

<https://royalsociety.org/journals/authors/author-guidelines/#supplementary-material> to include a suitable title and informative caption. An example of appropriate titling and captioning may be found at [https://figshare.com/articles/Table_S2_from_Is_there_a_trade-off_between_peak_performance_and_performance_breadth_across_temperatures_for_aerobic_sc](https://figshare.com/articles/Table_S2_from_Is_there_a_trade-off_between_peak_performance_and_performance_breadth_across_temperatures_for_aerobic_scope_in_teleost_fishes_/3843624) ope_in_teleost_fishes_/3843624.

Author's Response to Decision Letter for (RSOS-210406.R0)

See Appendix A.

RSOS-210406.R1 (Revision)

Review form: Reviewer 1

Is the manuscript scientifically sound in its present form?

Yes

Are the interpretations and conclusions justified by the results?

Yes

Is the language acceptable?

Yes

Do you have any ethical concerns with this paper?

No

Have you any concerns about statistical analyses in this paper?

No

Recommendation?

Accept as is

Comments to the Author(s)

Thanks for addressing my comments. I am glad that the authors found them useful. I do not have any further comments or questions.

Review form: Reviewer 2**Is the manuscript scientifically sound in its present form?**

Yes

Are the interpretations and conclusions justified by the results?

Yes

Is the language acceptable?

Yes

Do you have any ethical concerns with this paper?

No

Have you any concerns about statistical analyses in this paper?

No

Recommendation?

Major revision is needed (please make suggestions in comments)

Comments to the Author(s)

1. I regret, but I can still not follow the logic of the sunflower comparisons: "Although solar panels absorb light throughout the year, while sunflower inflorescences absorb light only in their 2-3-month growing season, the ideal azimuth of sunflower inflorescences and solar panels turns eastward."

(a) Young sunflowers track the Sun during the day. Hence, no ideal azimuth angle can be defined.

(b) Mature sunflowers don't track the Sun during the day, because they don't need to maximise their energy input anymore; they've already grown up; they just need enough light energy to stay alive. Hence, if mature sunflowers aren't in need to "absorb maximal light energy", what would the definition of their "ideal azimuth" angle reflect? How is "ideal" understood here?

(c) Solar panels must deliver the greatest harvesting efficiency over their entire lifetimes -- in contrast to sunflowers. Hence, ideally, they should track the Sun during the day -- like the young sunflowers. If this is not possible, ideally, the azimuth angle should guarantee the greatest harvest of the annual incoming solar radiation -- but this stands in contrast to mature sunflowers, c.f. (b).

Please clarify this aspect in the manuscript. I find the comparison of sunflowers, which only take as much energy as they need, with solar panels, which must harvest as much energy as only possible, rather more confusing than helpful. If these comparisons, therefore, aren't necessary for this study, why could they actually not all be dropped?

2. "Higher temperatures result in an increased dark-current reducing the power conversion efficiency, the effect of which is equivalent to that of the DECREASED irradiance of direct sunlight in the afternoon."

I respectfully disagree.

(a) Would the dark-current not decrease with decreased irradiance of direct sunlight?

(b) Higher temperatures increase the dark-current and the short-circuit current, c.f.

pveducation.org/pvcdrom/solar-cell-operation/effect-of-temperature. Hence, the effect of which is rather equivalent to that of a (slightly) INCREASED irradiance of direct sunlight in the afternoon.

3. Aerosol optical depth (AOD) is a measure of the extinction of the solar beam by dust and haze, i.e. by particles in the atmosphere (dust, smoke, pollution) that block sunlight by absorbing or by scattering light. How much solar energy passes through the atmospheric air mass, however, does also depend on the total precipitable water column, relative humidity, surface pressure, CO₂ concentration, and total-column abundance of ozone, etc. Therefore, connecting the temperature argument solely with the dust/aerosol concentration seems to disregard other important aspects of the atmospheric chemistry.

Higher temperatures reduce the conversion efficiency of PV panels as well as their potential energy yield; more/thicker clouds may reduce the energy yield, but they do not necessarily reduce the conversion efficiency of PV panels. Consequently, why should the effect of higher temperatures not have a greater impact on the azimuth angle? The present manuscript does not answer this question satisfactorily.

Decision letter (RSOS-210406.R1)

Dear Dr Horvath

The Editors assigned to your paper RSOS-210406.R1 "How the morning-afternoon cloudiness asymmetry affects the energy-maximizing azimuth direction of fixed-tilt monofacial solar panels" have made a decision based on their reading of the paper and any comments received from reviewers.

Regrettably, in view of the reports received, the manuscript has been rejected in its current form. However, a new manuscript may be submitted which takes into consideration these comments.

We invite you to respond to the comments supplied below and prepare a resubmission of your manuscript. Below the referees' and Editors' comments (where applicable) we provide additional requirements. We provide guidance below to help you prepare your revision.

[You will see that it is Reviewer 2 who continues to find the paper unsatisfactory. Any revision should address Reviewer 2's comments. We are likely to send a resubmitted paper back to Reviewer 2 and we may also send it to a new reviewer.]

Please note that resubmitting your manuscript does not guarantee eventual acceptance, and we do not generally allow multiple rounds of revision and resubmission, so we urge you to make every effort to fully address all of the comments at this stage. If deemed necessary by the Editors, your manuscript will be sent back to one or more of the original reviewers for assessment. If the original reviewers are not available, we may invite new reviewers.

Please resubmit your revised manuscript and required files (see below) no later than 25-May-2022. Note: the ScholarOne system will 'lock' if resubmission is attempted on or after this deadline. If you do not think you will be able to meet this deadline, please contact the editorial office immediately.

Please note article processing charges apply to papers accepted for publication in Royal Society Open Science (<https://royalsocietypublishing.org/rsos/charges>). Charges will also apply to papers transferred to the journal from other Royal Society Publishing journals, as well as papers submitted as part of our collaboration with the Royal Society of Chemistry (<https://royalsocietypublishing.org/rsos/chemistry>). Fee waivers are available but must be requested when you submit your manuscript (<https://royalsocietypublishing.org/rsos/waivers>).

Thank you for submitting your manuscript to Royal Society Open Science and we look forward to receiving your resubmission. If you have any questions at all, please do not hesitate to get in touch.

on behalf of Prof Peter Haynes (Subject Editor)
openscience@royalsociety.org

Associate Editor Comments to Author:

Thank you for the responses to reviewers. While one of the referees is now satisfied that your work can be accepted, there remain substantial questions to answer from the more critical reviewer. Given that we are not generally able to permit multiple rounds of revision, we are going to reject this iteration of the paper, but if you wish to substantially rework your study to respond to the reviewers' comments, you are welcome to do so.

Reviewer comments to Author:

Reviewer: 2

Comments to the Author(s)

1. I regret, but I can still not follow the logic of the sunflower comparisons: "Although solar panels absorb light throughout the year, while sunflower inflorescences absorb light only in their 2-3-month growing season, the ideal azimuth of sunflower inflorescences and solar panels turns eastward."

(a) Young sunflowers track the Sun during the day. Hence, no ideal azimuth angle can be defined.

(b) Mature sunflowers don't track the Sun during the day, because they don't need to maximise their energy input anymore; they've already grown up; they just need enough light energy to stay alive. Hence, if mature sunflowers aren't in need to "absorb maximal light energy", what would the definition of their "ideal azimuth" angle reflect? How is "ideal" understood here?

(c) Solar panels must deliver the greatest harvesting efficiency over their entire lifetimes -- in contrast to sunflowers. Hence, ideally, they should track the Sun during the day -- like the young sunflowers. If this is not possible, ideally, the azimuth angle should guarantee the greatest harvest of the annual incoming solar radiation -- but this stands in contrast to mature sunflowers, c.f. (b).

Please clarify this aspect in the manuscript. I find the comparison of sunflowers, which only take as much energy as they need, with solar panels, which must harvest as much energy as only possible, rather more confusing than helpful. If these comparisons, therefore, aren't necessary for this study, why could they actually not all be dropped?

2. "Higher temperatures result in an increased dark-current reducing the power conversion efficiency, the effect of which is equivalent to that of the DECREASED irradiance of direct sunlight in the afternoon."

I respectfully disagree.

(a) Would the dark-current not decrease with decreased irradiance of direct sunlight?

(b) Higher temperatures increase the dark-current and the short-circuit current, c.f.

pveducation.org/pvc/drom/solar-cell-operation/effect-of-temperature. Hence, the effect of which is rather equivalent to that of a (slightly) INCREASED irradiance of direct sunlight in the afternoon.

3. Aerosol optical depth (AOD) is a measure of the extinction of the solar beam by dust and haze, i.e. by particles in the atmosphere (dust, smoke, pollution) that block sunlight by absorbing or by scattering light. How much solar energy passes through the atmospheric air mass, however, does also depend on the total precipitable water column, relative humidity, surface pressure, CO₂ concentration, and total-column abundance of ozone, etc. Therefore, connecting the temperature argument solely with the dust/aerosol concentration seems to disregard other important aspects of the atmospheric chemistry.

Higher temperatures reduce the conversion efficiency of PV panels as well as their potential energy yield; more/thicker clouds may reduce the energy yield, but they do not necessarily reduce the conversion efficiency of PV panels. Consequently, why should the effect of higher temperatures not have a greater impact on the azimuth angle? The present manuscript does not answer this question satisfactorily.

Reviewer: 1

Comments to the Author(s)

Thanks for addressing my comments. I am glad that the authors found them useful. I do not have any further comments or questions.

===PREPARING YOUR MANUSCRIPT===

If you have been asked to revise the written English in your submission as a condition of publication, you must do so, and you are expected to provide evidence that you have received language editing support. The journal would prefer that you use a professional language editing service and provide a certificate of editing, but a signed letter from a colleague who is a fluent speaker of English is acceptable. Note the journal has arranged a number of discounts for authors using professional language editing services (<https://royalsociety.org/journals/authors/benefits/language-editing/>).

===PREPARING YOUR REVISION IN SCHOLARONE===

<https://royalsociety.org/journals/authors/author-guidelines/#data>. You should ensure that

you cite the dataset in your reference list. If you have deposited data etc in the Dryad repository, please include both the 'For publication' link and 'For review' link at this stage.

Author's Response to Decision Letter for (RSOS-210406.R1)

See Appendix B.

RSOS-211948.R0

Review form: Reviewer 2

Is the manuscript scientifically sound in its present form?

Yes

Are the interpretations and conclusions justified by the results?

Yes

Is the language acceptable?

Yes

Do you have any ethical concerns with this paper?

No

Have you any concerns about statistical analyses in this paper?

No

Recommendation?

Major revision is needed (please make suggestions in comments)

Comments to the Author(s)

A) Sunflower vs solar cells.

I respectfully disagree. The comparison between plants and solar panels is more distracting than helpful for the following reasons. Also, citations are often used to backup claims made in a

manuscript. Any manuscript, however, should stand for itself and be accessible for most interested readers, without studying any (outdated) citations first.

1. Fixed solar cells must maximise the insolation over a 365 day period, whereas sunflowers follow or see the sun over a much shorter time frame. The “ideal” azimuth angle for sunflowers is therefore defined over a very different time span than the ideal azimuth for solar panels. Hence, they’re not directly comparable, because they’ve different meanings.
2. As the flowers develop, they lose their flexibility of movement (for optimising their hourly insolation), such that the stems of mature sunflowers become stiffer and stationary (which might be optimised for seasonal insolation).
3. Many new varieties of sunflowers are bred so that the flower heads droop groundward as the plants mature. So birds cannot remove seeds as easily while the potential for diseases is reduced (caused by water collecting in the flower head). If such a downward tilted “head contributes more than 25% of the whole-plant light absorption at maturity”, a mature sunflower is probably not prioritising to maximise its insolation anymore.
4. Solar panels produce electrical energy; plants produce chemical energy. To which energy does the ideal azimuth refer? It is a little of comparing apples to oranges, <https://www.scientificamerican.com/article/plants-versus-photovoltaics-at-capturing-sunlight/>

Sunflowers produce energy not specifically from sunlight but through a chemical breakdown of bonds that hold molecules together. They also are not 100 percent reliant on sunshine for energy production. In fact, the efficiency of photosynthesis is less than 3% (and indeed plants are not black). They can use soil nutrients in conjunction with sunlight and water to make energy. That means they do not need as much sunlight since their recipe for energy is broader than that of a solar panel. Hence, their energy needs are also limited.

It is in this respect that I still have great difficulties to understand the comparison between the “energy-maximising” azimuth direction of sunflowers and solar panels. I recommend to drop the comparison between apples and oranges entirely.

B) Temperature vs cloud coverage.

“The effect of higher panel temperatures must have a smaller impact on the energy-maximizing ideal azimuth angle than the frequency of clouds, because industrial solar panel farms are predominantly installed in regions with minimal cloudiness.”

Following this train of thought, the quality of air must have a smaller impact on life expectancy than crossing the street by red, driving too fast or after a glass of wine, because many (adult) people are often ignoring the traffic lights, speed limits or BAC levels.

But even if the authors can provide data in support of their claim, e.g. cloudiness-index (its annual average) vs latitude, it would not be sufficient to explain the causation of a correlation. In deserts, as far as I understood, apart from the high temperatures, dust/sand accumulation on the solar panels is one of the greatest concerns. If so, the frequency of cleaning the panels will likely be more important than their azimuth angle.

Finally, colleagues at Tampere University looked into how cloud transitions affect the performance of real-world PV systems. For example, the irradiance incident on PV generators can considerably EXCEED the expected clear sky irradiance. Due to this phenomenon, called cloud enhancement (CE), the maximum power of the PV generator can exceed the rated power of the inverter connecting the generator to the grid, <https://doi.org/10.1063/5.0007550>. But more importantly, often the impact of cloud transitions on a PV system can simply be ignored (especially for large-scale systems), <http://dx.doi.org/10.1049/iet-rpg.2019.0085>, <https://doi.org/10.1016/j.renene.2020.01.119>.

Yes, cloud coverage will reduce the annual insolation (and thus PV solar yield) in overall, but “cloudy” countries are often also characterised by stronger winds and cooler temperatures, e.g. Ireland, while rain clouds actually help to keep solar panels clean and mostly free from debris. There is a reason for why solar farms are installed in the UK, too. Cloudy Norway is even further north, yet it is quite possible to produce solar energy there: Ås, a small town south of Oslo, receives 1000 kilowatt-hours (kWh) per square meter annually. This is comparable to many parts of Germany, where solar power has boomed over the last 10 years. Last but not least, if cloud coverage would be a greater impediment to solar PV installations than temperature, floating solar farms should have a darker and by far less brighter future.

In summary, the argument for why cloudiness (and the frequency of cloud transitions) should have a greater impact than temperature [on the optimum/ideal azimuth direction of solar panels] is here too weakly presented by the authors, since it requires a thoroughly elaborated discussion with numbers (data).

Decision letter (RSOS-211948.R0)

Dear Dr Horvath

The Editors assigned to your paper RSOS-211948 "How the morning-afternoon cloudiness asymmetry affects the energy-maximizing azimuth direction of fixed-tilt monofacial solar panels" have now received comments from reviewers and would like you to revise the paper in accordance with the reviewer comments and any comments from the Editors. Please note this decision does not guarantee eventual acceptance.

In short, the reviewer seems to have two major points -- (i) that the comparison with sunflowers is spurious and unhelpful and (ii) that temperature effects on solar cell productivity have not been taken properly into account. I find (i) reasonable. If you want to publish a paper in which the useful comparison between sunflowers and solar panels is the main point then you should do that, but more concrete evidence and argument would be needed -- you would need to submit a new paper. But my reading of the emphasis of the paper under consideration is that there may be some practical advantage to varying the orientation of solar panels from direct south/north if there is systematic am/pm asymmetry in cloudiness -- nothing is gained by the analogy with sunflowers.

I will be pleased to accept the paper if you remove the sunflower material (because it increases the length of the paper beyond what is justifiable from the content), if you make it absolutely clear, addressing the referee's point (ii) that the implications of any conclusions must be subject to further scrutiny re variation of efficiency with respect to temperature etc. I recommend the comment that seems to suggest that the fact that solar panel farms are (according to the authors) largely built in desert regions implies that cloudiness MUST outweigh temperature in important

in this respect -- the logic seems flawed and, again, the point does not seem important to conclusions of your paper.

I hope that the above recommendations provide a simple and reasonable approach to rapid publication of your paper in a form in which the major concrete conclusions remain as you intended. (If you made these changes then I would not see any reason to send the paper to the referee or referees once again.)

Please submit your revised manuscript and required files (see below) no later than 21 days from today's (ie 14-Feb-2022) date. Note: the ScholarOne system will 'lock' if submission of the revision is attempted 21 or more days after the deadline. If you do not think you will be able to meet this deadline please contact the editorial office immediately.

on behalf of Peter Haynes (Subject Editor)
openscience@royalsociety.org

Associate Editor Comments to Author:
Comments to the Author:

The reviewer has provided extensive commentary on your work (for which we're grateful), but it seems clear that there are a number of areas that need further modification and need to be addressed by you before the paper may be considered ready for publication.

When the revision is received, if the Editors consider that the paper has adequately addressed the reviewer's concerns (namely, that the sunflower comparison is not particularly relevant in this case and that the impact of temperature on solar cells has not been appropriately considered), then we will feel comfortable accepting the paper for publication - it will be for the wider research community to then engage with, comment on, and rebut/support the findings of the work.

If, however, concerns remain that the reviewer's concerns have not been adequately addressed, the paper may be rejected: in general, the journal only permits one round of major revision (and perhaps a further round of minor, largely presentational revisions). On receipt of your paper's next iteration, you will have had several rounds of revision to persuade the reviewers and Editors

that the paper is ready for acceptance, and it would not be fair on the reviewers (or Editors) to spend further time on reviewing the paper at that point.

We wish you luck and also look forward to receiving your response in due course.
Thanks.

Reviewer comments to Author:

Reviewer: 2

Comments to the Author(s)

A) Sunflower vs solar cells.

I respectfully disagree. The comparison between plants and solar panels is more distracting than helpful for the following reasons. Also, citations are often used to backup claims made in a manuscript. Any manuscript, however, should stand for itself and be accessible for most interested readers, without studying any (outdated) citations first.

1. Fixed solar cells must maximise the insolation over a 365 day period, whereas sunflowers follow or see the sun over a much shorter time frame. The "ideal" azimuth angle for sunflowers is therefore defined over a very different time span than the ideal azimuth for solar panels. Hence, they're not directly comparable, because they've different meanings.
2. As the flowers develop, they lose their flexibility of movement (for optimising their hourly insolation), such that the stems of mature sunflowers become stiffer and stationary (which might be optimised for seasonal insolation).
3. Many new varieties of sunflowers are bred so that the flower heads droop groundward as the plants mature. So birds cannot remove seeds as easily while the potential for diseases is reduced (caused by water collecting in the flower head). If such a downward tilted "head contributes more than 25% of the whole-plant light absorption at maturity", a mature sunflower is probably not prioritising to maximise its insolation anymore.
4. Solar panels produce electrical energy; plants produce chemical energy. To which energy does the ideal azimuth refer? It is a little of comparing apples to oranges,
<https://www.scientificamerican.com/article/plants-versus-photovoltaics-at-capturing-sunlight/>

Sunflowers produce energy not specifically from sunlight but through a chemical breakdown of bonds that hold molecules together. They also are not 100 percent reliant on sunshine for energy production. In fact, the efficiency of photosynthesis is less than 3% (and indeed plants are not black). They can use soil nutrients in conjunction with sunlight and water to make energy. That means they do not need as much sunlight since their recipe for energy is broader than that of a solar panel. Hence, their energy needs are also limited.

It is in this respect that I still have great difficulties to understand the comparison between the "energy-maximising" azimuth direction of sunflowers and solar panels. I recommend to drop the comparison between apples and oranges entirely.

B) Temperature vs cloud coverage.

"The effect of higher panel temperatures must have a smaller impact on the energy-maximizing ideal azimuth angle than the frequency of clouds, because industrial solar panel farms are predominantly installed in regions with minimal cloudiness."

Following this train of thought, the quality of air must have a smaller impact on life expectancy than crossing the street by red, driving too fast or after a glass of wine, because many (adult) people are often ignoring the traffic lights, speed limits or BAC levels.

But even if the authors can provide data in support of their claim, e.g. cloudiness-index (its annual average) vs latitude, it would not be sufficient to explain the causation of a correlation. In deserts, as far as I understood, apart from the high temperatures, dust/sand accumulation on the

solar panels is one of the greatest concerns. If so, the frequency of cleaning the panels will likely be more important than their azimuth angle.

Finally, colleagues at Tampere University looked into how cloud transitions affect the performance of real-world PV systems. For example, the irradiance incident on PV generators can considerably EXCEED the expected clear sky irradiance. Due to this phenomenon, called cloud enhancement (CE), the maximum power of the PV generator can exceed the rated power of the inverter connecting the generator to the grid, <https://doi.org/10.1063/5.0007550>. But more importantly, often the impact of cloud transitions on a PV system can simply be ignored (especially for large-scale systems), <http://dx.doi.org/10.1049/iet-rpg.2019.0085>, <https://doi.org/10.1016/j.renene.2020.01.119>.

Yes, cloud coverage will reduce the annual insolation (and thus PV solar yield) in overall, but “cloudy” countries are often also characterised by stronger winds and cooler temperatures, e.g. Ireland, while rain clouds actually help to keep solar panels clean and mostly free from debris. There is a reason for why solar farms are installed in the UK, too. Cloudy Norway is even further north, yet it is quite possible to produce solar energy there: Ås, a small town south of Oslo, receives 1000 kilowatt-hours (kWh) per square meter annually. This is comparable to many parts of Germany, where solar power has boomed over the last 10 years. Last but not least, if cloud coverage would be a greater impediment to solar PV installations than temperature, floating solar farms should have a darker and by far less brighter future.

In summary, the argument for why cloudiness (and the frequency of cloud transitions) should have a greater impact than temperature [on the optimum/ideal azimuth direction of solar panels] is here too weakly presented by the authors, since it requires a thoroughly elaborated discussion with numbers (data).

===PREPARING YOUR MANUSCRIPT===

Your revised paper should include the changes requested by the referees and Editors of your manuscript. You should provide two versions of this manuscript and both versions must be provided in an editable format:
 one version identifying all the changes that have been made (for instance, in coloured highlight, in bold text, or tracked changes);
 a 'clean' version of the new manuscript that incorporates the changes made, but does not highlight them. This version will be used for typesetting if your manuscript is accepted.

If you have been asked to revise the written English in your submission as a condition of publication, you must do so, and you are expected to provide evidence that you have received language editing support. The journal would prefer that you use a professional language editing

service and provide a certificate of editing, but a signed letter from a colleague who is a fluent speaker of English is acceptable. Note the journal has arranged a number of discounts for authors using professional language editing services (<https://royalsociety.org/journals/authors/benefits/language-editing/>).

===PREPARING YOUR REVISION IN SCHOLARONE===

<https://royalsociety.org/journals/authors/author-guidelines/#supplementary-material> to include a suitable title and informative caption. An example of appropriate titling and captioning

may be found at https://figshare.com/articles/Table_S2_from_Is_there_a_trade-off_between_peak_performance_and_performance_breadth_across_temperatures_for_aerobic_sc_ope_in_teleost_fishes_/3843624.

Author's Response to Decision Letter for (RSOS-211948.R0)

See Appendix C.

Decision letter (RSOS-211948.R1)

Dear Dr Horvath,

It is a pleasure to accept your manuscript entitled "How the morning-afternoon cloudiness asymmetry affects the energy-maximizing azimuth direction of fixed-tilt monofacial solar panels" in its current form for publication in Royal Society Open Science.

on behalf of Professor Peter Haynes (Subject Editor)
openscience@royalsociety.org

Appendix A

Point-by-Point Response to the Comments of Referee 1

We thank the positive and constructive review of Referee 1. Our manuscript was revised on the basis of the reports of two Reviewers. All changes suggested by Referee 1 and Referee 2 and performed by the Authors themselves are marked with green, orange and blue, respectively. Below is our detailed Point-by-Point Response to the comments of Referee 1.

Referee 1 wrote: *The paper explores the effect of morning-afternoon cloudiness asymmetry on the energy-maximizing azimuth direction of solar panels. The authors calculate the deviation of the energy maximizing azimuth from the standard south orientation which is typically used in the northern hemisphere. The paper is well-written, and the methodologies and results are discussed in sufficient detail. The topic addressed in the paper is novel and important from both practical and intellectual perspectives related to the understanding and deployment of solar photovoltaics systems. I have a few questions/comments:*

1) The analysis has been done on standard (monofacial) panels which are the dominant technology for today. Bifacial solar panels are however getting much attraction due to the reducing costs and potentially lower leveled cost of electricity for many locations of the world. See for example:

Carlos D. Rodríguez-Gallegos, Haohui Liu, Oktoviano Gandhi, Jai Prakash Singh, Vijay Krishnamurthy, Abhishek Kumar, Joshua S. Stein, Shitao Wang, Li Li, Thomas Reindl, Ian Marius Peters (2020) Global techno-economic performance of bifacial and tracking photovoltaic systems. Joule 4 (7): 1514-1541 (doi: 10.1016/j.joule.2020.05.005)

While it may require substantial additions to extend the computations done by the authors from the monofacial to bifacial, the authors may want to make a comment whether the same methodology could be applied for bifacial panels or not. Is there any qualitative insight they could provide for South faced fixed tilt bifacial panels with relatively simple extrapolation? A special case of vertically mounted East/West facing bifacial panels may be discussed relatively easily within the mathematical framework of the paper since this case may be approximated by a superposition of East facing monofacial (during mornings) and west faced monofacial (during afternoons).

2) Related to the previous comment, a similar comment may be relevant for the single axis tracking panels which usually track the direct beam around the East/West axis. For morning/evening asymmetry scenario, an intelligent tracking may be implemented to maximize energy [see for example: Patel, M. Tahir; Ahmed, M. Sojib; Imran, Hassan; Butt, Nauman Z.; Khan, M. Ryyan; Alam, Muhammad A. (2021) Global analysis of next-generation utility-scale PV: Tracking bifacial solar farms. Applied Energy vol. 290: 116478, doi: 10.1016/j.apenergy.2021.116478]. Authors may want to refer this paper to broaden the readers' perspective.

Answer: The title was changed as follows: **How the morning-afternoon cloudiness asymmetry affects the energy-maximizing azimuth direction of fixed-tilt monofacial solar panels**

Furthermore, we used frequently the indicator 'monofacial' for the studied fixed-tilt solar panels throughout the revised manuscript.

To the revised Discussion we added the following two paragraphs:

The current analysis focused on standard fixed-tilt monofacial solar panels, which are the dominant technology today. Bifacial solar panels are, however, getting increasing attention due to the potentially lower cost of electricity for many locations of the world [10]. The computational methodology used in this work can also be applied to bifacial panels. Such a detailed analysis, which should take into consideration the absorption of earthlight (i.e. sky- and sunlight reflected from the ground) by the panel's rear side, is the topic of future research. Here we only mention the qualitative expectation that depending on the tilt angle, the energy-maximizing ideal azimuth of fixed-tilt bifacial panels also turns from the conventional geographical south in regions where afternoons are usually cloudier than mornings. The rear side of fixed-tilt bifacial panels with front side facing approximately south or southeast receives practically only diffuse skylight and earthlight, the irradiance of which is an order of magnitude less than that of direct sunlight. Thus, the ideal azimuth of such panels is also predominantly determined by sunlight and therefore similar to the azimuth of fixed-tilt monofacial panels.

In order to broaden the perspective of our analysis, it is worth performing similar computations for single axis monofacial and bifacial tracking panels, which usually track the direct sunlight around the east-west axis [26]. For morning/evening cloudiness asymmetry scenarios, an intelligent tracking could be implemented to maximize the absorbed energy.

10. Rodríguez-Gallegos CD, Liu H, Gandhi O, Singh JP, Krishnamurthy V, Kumar A, Stein JS, Wang S, Li L, Reindl T, Peters IM. 2020 Global techno-economic performance of bifacial and tracking photovoltaic systems. *Joule* **4**, 1514-1541.
26. Patel MT, Ahmed MS, Imran H, Butt NZ, Khan MR, Alam MA. 2021 Global analysis of next-generation utility-scale PV: Tracking bifacial solar farms. *Applied Energy* **290**, 116478. (doi: 10.1016/j.apenergy.2021.116478)

Referee 1 wrote: *3) The authors state that the temperature and dust effects could be assumed symmetric for morning-afternoons. I wonder whether the cloudiness asymmetry for morning-evening has any effect on temperature? A common perception is that the cloudy periods during the daytime are relatively lower in temperature compared to the otherwise sunny conditions. Can the authors verify the annual symmetry for temperature and aerosol particulate matter through the typical meteorological data for the locations explored in the paper?*

Answer: To the revised Discussion we added the following:

In the current computations the effects of (i) atmospheric dust and aerosol as well as (ii) the temperature of the solar panel on the photovoltaic efficiency were neglected. Over most urban/industrial sites, the aerosol optical depth increases by 10-40 % during the day with a maximum aerosol loading in the afternoon, as revealed by ground-based measurements from the Aerosol Robotic Network (AERONET) [27]. Thus, under typical clear-sky conditions, the irradiance of direct sunlight is slightly lower and the irradiance of diffuse (aerosol-scattered) skylight is slightly higher in the afternoon than in the morning. This aerosol-induced asymmetry in morning-afternoon illumination is equivalent to the asymmetry caused by the diurnal cycle of cloudiness. At a few sites, however, local meteorology (e.g. evening sea breeze) can result in a decreasing aerosol loading during the day. A further improvement of our model can incorporate AERONET measurements to account for such atypical local conditions.

Since solar cells are encapsulated in photovoltaic modules, their operating temperature is usually higher than the ambient air temperature. This is especially the case in the afternoon, when the air temperature is higher and more heat is radiated out by the ground than in the morning. Higher temperatures result in an increased dark-current reducing the power conversion efficiency [1, 5, 8], the effect of which is equivalent to that of the decreased irradiance of direct sunlight in the afternoon.

Therefore, the diurnal effects of dust/aerosol concentration and those of temperature on photovoltaic efficiency would typically lead to an even larger eastward deviation of the energy-maximizing ideal azimuth from the conventional southern direction; as if the afternoons were cloudier than assumed in our computations. Thus, the turn of the ideal azimuth is likely underestimated in this work.

In sum, the ideal azimuth angle is influenced by the asymmetric morning-afternoon dust/aerosol concentration and ambient temperature similarly to the asymmetric morning-afternoon cloudiness. As a result, the energy-maximizing azimuth of fixed-tilt monofacial solar panels deviates from south, if the annual-average cloudiness of mornings and afternoons differs.

27. Smirnov A, Holben BN, Eck TF, Slutsker I, Chatenet B, Pinker RT. 2002 Diurnal variability of aerosol optical depth observed at AERONET (Aerosol Robotic Network) sites. *Geophysical Research Letters* **29**, 2115. (doi: 10.1029/2002GL016305)

Point-by-Point Response to the Comments of Referee 2

We thank the constructive review of Referee 2. Our manuscript was revised on the basis of the reports of two Reviewers. All changes suggested by Referee 1 and Referee 2 and performed by the Authors themselves are marked with green, orange and blue, respectively. Below is our detailed Point-by-Point Response to the comments of Referee 2.

Referee 2 wrote: 1. "Fixed-tilt solar panels conventionally face south." That's not true in Australia, where they're facing north (and towards the Equator). Please correct.

Answer: In the revised manuscript this was corrected as follows:

Abstract:

In the northern hemisphere, south is the conventional azimuth direction of fixed-tilt monofacial solar panels, because this orientation may maximize the received light energy.

Introduction:

On the northern/southern hemisphere of the Earth, fixed-tilt monofacial solar panels conventionally face south/north, because the southern/northern azimuth may ensure maximal solar energy [1, 2, 3, 4, 5, 6, 7, 8, 9].

Referee 2 wrote: 2. "Mature sunflower inflorescences absorb maximal light energy, if they face east". Yes, the mature sunflower is not heliotropic. Yet, sunflowers do track the sun during the day from east to west at the bud stage, i.e. when they are in need of great energy resources. Please incorporate this important detail in your analysis.

Answer: As requested, to the revised Introduction we added the following:

On the other hand, the inflorescences of non-heliotropic mature sunflowers (*Helianthus annuus*) face east. Only young sunflowers do track the sun during the day from east to west at their bud stage, i.e. when they are in need of great energy resources.

Referee 2 wrote: 3. Reference [1] is quoted, but Ref [1] uses a range from $\lambda_{\min} = 170 \text{ nm}$ to $\lambda_{\max} = 10 \text{ um}$, contradicting the here selected "relevant wavelength interval of sky radiation". Actually, $\lambda_{\min} = 250 \text{ nm}$ is often not relevant, since light with $\lambda < 380 \text{ nm}$ is typically absorbed by the glass/encapsulant layers. In addition, for silicon solar cells $\lambda_{\max} = 1200 \text{ nm} \gg 900 \text{ nm}$. In conclusion, the authors seem to have not only opted for a too narrow bandwidth interval by including the often irrelevant UV range (250 nm - 380 nm). How will the results change for a more appropriate spectral range, e.g. from 400 nm to 1200 nm?

Answer: On the one hand, the criticized sentence was corrected as follows:

where $\tau_{\text{diff}}(\theta_n)$ is the net transmissivity of the panel's dielectric layer for diffuse skylight, $I_{\text{diff}}(\lambda, t)$ is the diffuse irradiance received by a horizontal surface, and $\lambda_{\min} = 200 \text{ nm} \leq \lambda \leq \lambda_{\max} = 4000 \text{ nm}$ is the solar-energetically relevant wavelength interval of sky radiation [1].

One the other hand, we calculated (see subsections 2.3 and 2.4) the maximum light energy available for Fresnel-reflecting ($R > 0$) and anti-reflective ($R = 0$) fixed-tilt monofacial solar panels. For this we used the mean power flux W_{Sun} (W/m^2) of direct sunlight and the mean power flux W_{diff} (W/m^2) of diffuse skylight averaged for the period 2009-2019 and measured by a horizontal detector surface of ERA5 – ECMWF with $\lambda_{\text{min}} = 200 \text{ nm} \leq \lambda \leq \lambda_{\text{max}} = 4000 \text{ nm}$ [17]. In the revised manuscript we wrote:

Two products are evaluated in the present study for the wavelength range $\lambda_{\text{min}} = 200 \text{ nm} \leq \lambda \leq \lambda_{\text{max}} = 4000 \text{ nm}$

Thus, our results does not change for the spectral range mentioned by Referee 2.

Referee 2 wrote: 4. Why does the absorption spectrum A in Eq. 8 not depend on the angle of incidence (AOI)? The factor $1-R$ accounts for the external reflection losses, but the escape reflection losses do greatly depend on the absorber thickness and the AOI. Please clarify.

5. Since $A(\lambda)$ is here independent of the AOI, the study solely focuses on the external reflection losses and thus underestimates the total reflection, i.e. it overestimates the total absorption. Despite this point becomes irrelevant, because the authors use $A = 1$ later on, it should be made clear in the text that only external reflection losses were considered in the calculations.

Answer: To the 2.2. subsection of the Materials and Methods we added:

The smooth, Fresnel-reflecting dielectric with reflectivity $R(\cos\gamma)$ transmits $1 - R(\cos\gamma)$ proportion of the incident light towards the underlying absorber layer, the absorption spectrum of which is $0 \leq A(\lambda, \gamma) \leq 1$, where γ is the incidence angle from the normal vector of the surface. Thus, the net absorbance of the solar panel is:

$$A_{\text{net}} = [1 - R(\cos\gamma)] \cdot A(\lambda, \gamma). \quad (8)$$

Later on (see subsection 2.3.) we consider only the case $A(\lambda, \gamma) = 1$, because we calculate the maximal possible total light energy per unit area available for a Fresnel-reflecting ($R > 0$) fixed-tilt solar panel integrated for the whole year.

Referee 2 wrote: 6. Please convert MJ/cm^2 into $\text{kWh}/\text{year}/\text{cm}^2$.

Answer: $1 \text{ MJ} = 10^6 \text{ J}$ and $1 \text{ kWh} = 10^3 \text{ J/s} \cdot 3600 \text{ s} = 3.6 \cdot 10^6 \text{ J}$, thus $1 \text{ kWh} = 3.6 \text{ MJ}$ and $1 \text{ MJ} = (1/3.6) \text{ kWh} \approx 0.278 \text{ kWh}$. Using the conversion $1 \text{ MJ} \approx 0.278 \text{ kWh}$, in the revised Figures 5 and 6 the total energy e absorbed by a unit-surface (1 m^2) solar panel in one year is given not only in $\text{MJ}/\text{m}^2/\text{year}$, but also in $\text{kWh}/\text{m}^2/\text{year}$:

Figure 5. Colour-coded values of the total light energy e (in $\text{MJ/m}^2/\text{year}$ as well as $\text{kWh/m}^2/\text{year}$) per unit area available for a Fresnel-reflecting (reflectivity $R > 0$) fixed-tilt monofacial solar panel between 1 January and 31 December in A) Boone County (39.0° N , -84.75° E), B) Tennessee (35.5° N , -88.25° E), C) Georgia (31.25° N , -83.25° E), D) Central Italy (41.0° N , 15.0° E), E) Central Hungary (47.0° N , 19.0° E) and F) South Sweden (58.0° N , 13.0° E), as functions of the elevation angle θ_n (from the horizontal) and the azimuth angle α_n (clockwise from the geographical south) of the panel's normal vector. The Fresnel's reflectivity $R(\lambda) > 0$ of the smooth outer surface of the solar panel is shown in Fig. 3B. The black continuous curves mark the ideal (θ_n^*, α_n^*) angle pairs for which e is maximal for a given θ_n^* .

Figure 6. Same as Fig. 5 but for an anti-reflective solar panel with zero reflectivity $R(\lambda) = 0$.

Referee 2 wrote: 7a. As solar cells are encapsulated in PV modules, their operating temperature is in general higher than the ambient temperature. This is especially the case in the afternoon, when more heat is radiated out by Earth's surface, once the local insolation has passed its peak value. However, higher temperatures can lead to significant increases in the dark-current and in turn to a reduction in the power conversion efficiency. How do you separate the impact of morning-afternoon ambient temperature from the morning-afternoon cloudiness?

7b. You state that the effect of temperature does not affect your main conclusions, because "in yearly average the neglected [temperature] effects influence the morning and afternoon photovoltaic efficiencies equally." Without numbers, this statement is a speculation and does not justify the neglect of temperature dynamics. More important of whether two effects influence the PV efficiencies equally or unequally is to quantify/estimate the magnitude of these two effects. Hence, what influences the azimuth angle the most? Is it the morning-afternoon ambient temperature, or is it the morning-afternoon cloudiness?

Answer: To the revised Discussion we added the following:

In the current computations the effects of (i) atmospheric dust and aerosol as well as (ii) the temperature of the solar panel on the photovoltaic efficiency were neglected. Over most urban/industrial sites, the aerosol optical depth increases by 10-40 % during the day with a maximum aerosol loading in the afternoon, as revealed by ground-based measurements from the Aerosol Robotic Network (AERONET)

[27]. Thus, under typical clear-sky conditions, the irradiance of direct sunlight is slightly lower and the irradiance of diffuse (aerosol-scattered) skylight is slightly higher in the afternoon than in the morning. This aerosol-induced asymmetry in morning-afternoon illumination is equivalent to the asymmetry caused by the diurnal cycle of cloudiness. At a few sites, however, local meteorology (e.g. evening sea breeze) can result in a decreasing aerosol loading during the day. A further improvement of our model can incorporate AERONET measurements to account for such atypical local conditions.

Since solar cells are encapsulated in photovoltaic modules, their operating temperature is usually higher than the ambient air temperature. This is especially the case in the afternoon, when the air temperature is higher and more heat is radiated out by the ground than in the morning. Higher temperatures result in an increased dark-current reducing the power conversion efficiency [1, 5, 8], the effect of which is equivalent to that of the decreased irradiance of direct sunlight in the afternoon.

Therefore, the diurnal effects of dust/aerosol concentration and those of temperature on photovoltaic efficiency would typically lead to an even larger eastward deviation of the energy-maximizing ideal azimuth from the conventional southern direction; as if the afternoons were cloudier than assumed in our computations. Thus, the turn of the ideal azimuth is likely underestimated in this work.

In sum, the ideal azimuth angle is influenced by the asymmetric morning-afternoon dust/aerosol concentration and ambient temperature similarly to the asymmetric morning-afternoon cloudiness. As a result, the energy-maximizing azimuth of fixed-tilt monofacial solar panels deviates from south, if the annual-average cloudiness of mornings and afternoons differs.

27. Smirnov A, Holben BN, Eck TF, Slutsker I, Chatenet B, Pinker RT. 2002 Diurnal variability of aerosol optical depth observed at AERONET (Aerosol Robotic Network) sites. *Geophysical Research Letters* **29**, 2115. (doi: 10.1029/2002GL016305)

Appendix B

Point-by-Point Response to the 2nd Review of Referee 2

We thank the 2nd review of Referee 2. Our manuscript was revised on the basis of this review. All changes suggested by Referee 2 are marked with orange. Below is our detailed Point-by-Point Response to the comments of Referee 2.

Referee 2 wrote: *1. I regret, but I can still not follow the logic of the sunflower comparisons:*

"Although solar panels absorb light throughout the year, while sunflower inflorescences absorb light only in their 2-3-month growing season, the ideal azimuth of sunflower inflorescences and solar panels turns eastward."

(a) Young sunflowers track the Sun during the day. Hence, no ideal azimuth angle can be defined.

(b) Mature sunflowers don't track the Sun during the day, because they don't need to maximise their energy input anymore; they've already grown up; they just need enough light energy to stay alive. Hence, if mature sunflowers aren't in need to "absorb maximal light energy", what would the definition of their "ideal azimuth" angle reflect? How is "ideal" understood here?

(c) Solar panels must deliver the greatest harvesting efficiency over their entire lifetimes -- in contrast to sunflowers. Hence, ideally, they should track the Sun during the day -- like the young sunflowers. If this is not possible, ideally, the azimuth angle should guarantee the greatest harvest of the annual incoming solar radiation -- but this stands in contrast to mature sunflowers, c.f. (b).

Please clarify this aspect in the manuscript. I find the comparison of sunflowers, which only take as much energy as they need, with solar panels, which must harvest as much energy as only possible, rather more confusing than helpful. If these comparisons, therefore, aren't necessary for this study, why could they actually not all be dropped?

Answer: Considering the biology of sunflowers (*Helianthus annuus*), we suggest Referee 2 to consult the extended literature of *Helianthus annuus*, including our paper:

Horváth G, Slíz-Balogh J, Horváth Á, Egri Á, Virágh B, Horváth D, Jánosi IM (2020) Sunflower inflorescences absorb maximum light energy if they face east and afternoons are cloudier than mornings. *Scientific Reports* 10: 21597 (doi: 10.1038/s41598-020-78243-z)

Ad 1) In our present manuscript, we did not state at all that young sunflowers (the leaves and immature head of which still track the Sun during the day) would have any ideal (i.e. energy-maximizing) azimuth angle.

Ad 2) It is incorrect to say that mature sunflowers (which no longer track the Sun) do not need to maximise their energy input anymore, since they have already grown up and therefore they just need enough light energy to stay alive. Contrary to this, the leaves of sunflowers continue sun-tracking with a dampened amplitude [Shell et al. 1974] after the azimuth direction of their mature inflorescences becomes fixed toward east after anthesis (the flowering period, especially the maturing of the stamens), because the whole plant needs henceforward light energy for its physiological processes and the development of the growing head.

Shell GSG, Lang ARG, Sale PJM (1974) Quantitative measures of leaf orientation and heliotropic response in sunflower, bean, pepper and cucumber. *Agricultural Meteorology* 13: 25-37

Ad 3) It is incorrect to say that mature sunflowers are not in need of absorbing maximal light energy. Contrary to this, the rapidly growing mature inflorescences need to absorb maximal light energy after

anthesis for the development of their blossoms and seeds. The head contributes more than 25% of the whole-plant light absorption at maturity [Rey et al. 2008]. Horváth et al. (2020) showed that mature sunflower inflorescences facing the geographical east absorb maximal light energy, if afternoons are usually cloudier than mornings.

Rey H, Dauzat J, Chenu K, Barczi JF, Dosio GAA, Lecoeur J (2008) Using a 3-D virtual sunflower to simulate light capture at organ, plant and plot levels: Contribution of organ interception, impact of heliotropism and analysis of genotypic differences. *Annals of Botany* 101: 1139-1151

Ad 4) The definition of the ‘ideal azimuth’ of mature sunflower inflorescences is that they absorb maximal light energy if they face this ideal direction.

Ad 5) On the basis of points 1)-4), it is incorrect to say that mature sunflower inflorescences do not need to deliver the greatest absorbed light energy (harvesting efficiency) over their entire lifetimes between anthesis and senescence. In fact, the ideal azimuth angle of mature sunflower inflorescences guarantees the greatest harvest of the incoming solar and sky radiation in the growing season. Hence, in this respect there is no contrast between mature sunflower inflorescences and fixed-tilt solar panels.

Ad 6) Finally, we call the attention of Referee 2 to the fact that the comparison between sunflowers and fixed-tilt solar panels is valid specifically for the non-heliotropic mature sunflower inflorescences, rather than for the whole plant including heliotropic leaves and immature bud.

Based on the above, we kept the comparison of sunflowers (the mature inflorescences of which must take as much light energy as possible during their growing season) with solar panels (which must harvest as much energy as possible throughout the year), because we consider this comparison helpful. The criticized paragraphs were revised in the Introduction and Discussion as follows:

Introduction

On the other hand, after anthesis the non-heliotropic (i.e. tracking no longer the Sun) mature inflorescences of sunflowers (*Helianthus annuus*) face east. Only young sunflowers do track the sun during the day from east to west at their bud stage. The leaves of sunflowers continue sun-tracking to a dampened extent [11] after the azimuth direction of the mature inflorescences becomes fixed toward east, because the whole plant needs henceforward light energy for its physiological processes and the development of the growing head. After anthesis, the rapidly growing mature inflorescences need to absorb maximal light energy for the development of their blossoms and seeds. The head contributes more than 25% of the whole-plant light absorption at maturity [12]. Using an atmospheric radiation model with measured cloudiness and plant-physiological input data, Horváth *et al.* [13] showed that mature sunflower inflorescences absorb maximal light energy, if they face geographical east and the afternoons are usually cloudier than the mornings in summer, as is the case in the area from which domesticated sunflowers originate [14]. Thus, ‘ideal azimuth angle’ of mature sunflower inflorescences means that they absorb maximal light energy if facing this azimuth direction.

There is some similarity between constantly east-facing mature sunflower inflorescences and fixed-tilt monofacial solar panels: both absorb as much light energy as possible with a fixed azimuth direction. However, there are three main differences between them:

- The absorption spectra of sunflower inflorescences and solar panels are different.
- Solar panels absorb sun/skylight throughout the year, while mature sunflower inflorescences absorb light only in their 2-3-month summer growing season between anthesis and senescence.

- The average elevation angle θ of the normal vector of mature sunflower inflorescences relative to the horizontal gradually decreases from about $+10^\circ$ to -75° during their growing season [13], while the normal vector of fixed-tilt monofacial solar panels has a constant ideal elevation angle $\theta \geq +45^\circ$, depending mainly on latitude [1].

Can these differences explain the large difference between the energy-maximizing (i.e. ideal) eastern azimuth direction of mature sunflower inflorescences and the southern azimuth of fixed-tilt monofacial solar panels? How does the morning-afternoon cloudiness asymmetry affect the energy-maximizing azimuth of such solar panels?

Discussion

Interestingly, there is some similarity between the energy-maximizing ideal azimuth direction of fixed-tilt monofacial solar panels and that of non-heliotropic (i.e. non-sun-tracking) mature sunflower inflorescences: in both cases the ideal azimuth turns eastward, if afternoons are cloudier than mornings. This similarity exists despite the different absorption spectra, elevation angles, and activity period of sunflower inflorescences and solar panels. Mature sunflower inflorescences absorb light only in the summer months, during the flowering period between anthesis and senescence, and their ideal azimuth turns eastward. Depending on the regional cloud conditions, mature sunflower inflorescences facing east receive 54-77 % more energy than those facing south, if afternoons are generally cloudier than mornings, as is typical in the cultivation regions of sunflowers. This excess light energy is an obvious ecological advantage of east facing compared to south facing. Note that the analogy between sunflowers and fixed-tilt monofacial solar panels is valid only for the non-heliotropic mature sunflower inflorescences, rather than for the whole plant including the heliotropic (sun-tracking) leaves and immature bud.

Referee 2 wrote: 2. "Higher temperatures result in an increased dark-current reducing the power conversion efficiency, the effect of which is equivalent to that of the DECREASED irradiance of direct sunlight in the afternoon."

I respectfully disagree.

(a) Would the dark-current not decrease with decreased irradiance of direct sunlight?

(b) Higher temperatures increase the dark-current and the short-circuit current, c.f. pveducation.org/pvcdrom/solar-cell-operation/effect-of-temperature. Hence, the effect of which is rather equivalent to that of a (slightly) INCREASED irradiance of direct sunlight in the afternoon.

Higher temperatures reduce the conversion efficiency of PV panels as well as their potential energy yield; more/thicker clouds may reduce the energy yield, but they do not necessarily reduce the conversion efficiency of PV panels. Consequently, why should the effect of higher temperatures not have a greater impact on the azimuth angle? The present manuscript does not answer this question satisfactorily.

Answer: In our 2nd revised paper, the paragraphs discussing the possible effect of the operating temperature of a fixed-tilt monofacial solar panel to its energy-maximizing ideal azimuth direction were revised as follows:

Second, the temperature dependence of a solar cell's power generation efficiency is also neglected in the current calculations. This conversion efficiency decreases/increases by 0.2-0.5 % for every 1 °C increase/decrease in temperature above/below the 25 °C reference temperature used in standard test conditions [31]. The operational cell temperature is higher than the ambient air temperature during daytime and primarily depends on the thermal properties of the cell material, the geometry and orientation

of the panel, the type of the background surface (roof, wall, or open field), the solar insolation, and the amount of ventilation, which in turn depends on wind speed. Therefore, the actual temperature variation is highly location and installation specific. Nevertheless, both weather data-based thermal modelling studies and long-duration outdoor tests indicate that the diurnal cycle of cell temperature is usually skewed towards the afternoon, even in cloudy conditions: that is, the cell temperature is generally higher in the afternoon than in the morning [32, 33, 34]. The resulting (opposite) asymmetry in conversion efficiency, similar to the asymmetry in cloudiness and aerosol load, favours the morning, that is, the eastern hemisphere.

Because industrial solar panel farms are predominantly installed in regions with minimal cloudiness (e.g. deserts), the effect of higher panel temperatures must have a smaller impact on the energy-maximizing ideal azimuth angle than the frequency of clouds: The panel's energy gain G due to the more intense direct solar radiation (in the absence of clouds) is greater than its energy loss L due to the smaller power conversion efficiency at higher temperatures ($G > L$). Otherwise (if $G < L$), industrial-scale solar panels would mainly be installed in regions with frequent clouds, which is not the case.

Taken together, the published observational data on the typical diurnal cycle of aerosol load and solar panel operating temperature as well as our ERA5-based radiation calculations strongly suggest the eastward turn of the ideal, energy-maximizing azimuth from due south, at locations where mornings are less cloudy than afternoons. The ideal azimuth direction can be further refined, and the expected eastward turn confirmed, if site-specific weather data are available at higher spatio-temporal resolutions than the ones provided by the global atmospheric reanalysis used in the current work.

Referee 2 wrote: *3. Aerosol optical depth (AOD) is a measure of the extinction of the solar beam by dust and haze, i.e. by particles in the atmosphere (dust, smoke, pollution) that block sunlight by absorbing or by scattering light. How much solar energy passes through the atmospheric air mass, however, does also depend on the total precipitable water column, relative humidity, surface pressure, CO₂ concentration, and total-column abundance of ozone, etc. Therefore, connecting the temperature argument solely with the dust/aerosol concentration seems to disregard other important aspects of the atmospheric chemistry.*

Answer: This criticism is irrelevant in the case of our model, because all these aspects are taken into consideration in the used ERA5 radiation quantities. In fact, not only the asymmetric morning-afternoon cloudiness and dust/aerosol concentration, but also many other important aspects of the atmospheric chemistry are incorporated in our model computations. In the revised subsection '2.3. ERA5 radiation data' and Discussion we clarified:

In the ERA5 radiation scheme, incoming solar radiation is attenuated by absorbing gases (water vapour, carbon dioxide, methane, ozone, other trace gases) and is scattered by molecules, aerosols, and cloud particles [20]. For water vapour and clouds, the radiation scheme uses prognostic information from the forecast model. For ozone, only diagnostic values are used (i.e. ozone has no feedback on the atmosphere via the radiation scheme); however, ozone profiles, total column ozone estimates, and ozone-sensitive channel radiances from a large number of sub-daily satellite observations are assimilated in the reanalysis. The spatial and seasonal distribution of greenhouse gases (CO₂, CH₄, N₂O, CFC-11, CFC-12) are prescribed by monthly zonally-averaged concentration profiles. The blocking of solar radiation by aerosols is described by climatological distributions of optical depth from sea salt, soil/dust, black carbon, and sulphate (including stratospheric sulphate from major volcanic eruptions of the last century). Input are monthly mean geographical profiles of optical depth, which account for large-scale seasonal variations. The contribution of local diurnal variations in aerosol optical depth, which is the only major radiative effect missing from ERA5, is discussed in section 4.

As for the reliability of ERA5 radiation data, there are some (mostly local) validations and intercomparisons with other reanalyses. One of the most comprehensive recent reviews by Yang and Bright [21] compared 6 new generation satellite derived data sets and two reanalyses, ERA5 and MERRA-2 (Modern-Era Retrospective analysis for Research and Applications, Version 2) with 27 years of continuous terrestrial observations on 57 reference sites, with hourly resolution. Satellite data are difficult to compare with reanalyses (they provide neither spatial nor temporal global coverage), but the final conclusion of [21] is that ERA5 clearly outperforms MERRA-2. More restricted regional comparisons have very similar conclusions, e.g. over the Indonesian region [22]. Two recent validations using Chinese records observed larger errors; however, they noted that cloudy-rainy regions showed the largest deviations, which is the consequence of **the relatively poor representation of clouds in all global weather forecast and climate models [23, 24]**. Overall, ERA5 currently represents the most accurate global description of the state of the atmosphere.

Discussion

Finally, we need to consider the potential effects on our results of the two main limitations of the current study. First, although the ERA5 radiation calculations do account for the large-scale (geographic) and low-frequency (monthly) variability of aerosols, they neglect the local diurnal variation of aerosol loading, which mainly affects the direct solar component. Over most urban/industrial sites, the aerosol optical depth increases by 10-40 % during the day with a maximum in the afternoon, as revealed by ground-based measurements from the Aerosol Robotic Network (AERONET) [30]. Thus, the irradiance of the dominant direct sunlight is slightly lower and the irradiance of the diffuse (aerosol-scattered) skylight is slightly higher in the afternoon than in the morning. This aerosol-induced asymmetry in morning-afternoon illumination is analogous to the asymmetry caused by the diurnal cycle of cloudiness and turns the ideal azimuth further east at most locations. At a few sites, however, local meteorology (e.g. afternoon sea breeze) can result in a decreasing aerosol loading during the day. An improved model could incorporate AERONET measurements to quantify the added eastward azimuth turn under typical conditions but also to account for atypical aerosol loads.

Appendix C

Dear Editorial Office of *Royal Society Open Science*,
and
Prof. Peter Haynes (Subject Editor),

thank you for your letter of 14 February 2022 in which you sent me the 3rd report of Referee 2 about our paper RSOS-211948 (= RSOS-210406.R2 = 2nd revised manuscript) entitled

How the morning-afternoon cloudiness asymmetry affects the energy-maximizing azimuth direction of fixed-tilt monofacial solar panels

by Peter Takacs, Judit Sliz-Balogh, Akos Horvath, Daniel Horvath,
Imre M. Janosi and Gabor Horvath

Subject Editor, Peter Haynes wrote: *In short, the reviewer seems to have two major points: (i) that the comparison with sunflowers is spurious and unhelpful and (ii) that temperature effects on solar cell productivity have not been taken properly into account. I find (i) reasonable. If you want to publish a paper in which the useful comparison between sunflowers and solar panels is the main point then you should do that, but more concrete evidence and argument would be needed -- you would need to submit a new paper. But my reading of the emphasis of the paper under consideration is that there may be some practical advantage to varying the orientation of solar panels from direct south/north if there is systematic am/pm asymmetry in cloudiness -- nothing is gained by the analogy with sunflowers.*

I will be pleased to accept the paper if you remove the sunflower material (because it increases the length of the paper beyond what is justifiable from the content), if you make it absolutely clear, addressing the referee's point (ii) that the implications of any conclusions must be subject to further scrutiny re variation of efficiency with respect to temperature etc.

Answer: As suggested by Referee 2 and the Editor, in our 3rd revised manuscript we deleted the following **comparison between sunflowers and solar panels**:

Introduction:

On the other hand, after anthesis the non-heliotropic (i.e. tracking no longer the Sun) mature inflorescences of sunflowers (*Helianthus annuus*) face east. Only young sunflowers do track the sun during the day from east to west at their bud stage. The leaves of sunflowers continue sun-tracking to a dampened extent [11] after the azimuth direction of the mature inflorescences becomes fixed toward east, because the whole plant needs henceforward light energy for its physiological processes and the development of the growing head. After anthesis, the rapidly growing mature inflorescences need to absorb maximal light energy for the development of their blossoms and seeds. The head contributes more than 25% of the whole-plant light absorption at maturity [12]. Using an atmospheric radiation model with measured cloudiness and plant-physiological input data, Horváth *et al.* [13] showed that mature sunflower inflorescences absorb maximal light energy, if they face geographical east and the afternoons are usually cloudier than the mornings in summer, as is the case in the area from which domesticated sunflowers originate [14]. Thus, 'ideal azimuth angle' of mature sunflower inflorescences means that they absorb maximal light energy if facing this azimuth direction.

There is some similarity between constantly east-facing mature sunflower inflorescences and fixed-tilt monofacial solar panels: both absorb as much light energy as possible with a fixed azimuth direction. However, there are three main differences between them:

- The absorption spectra of sunflower inflorescences and solar panels are different.

- Solar panels absorb sun/skylight throughout the year, while mature sunflower inflorescences absorb light only in their 2-3-month summer growing season between anthesis and senescence.
- The average elevation angle θ of the normal vector of mature sunflower inflorescences relative to the horizontal gradually decreases from about $+10^\circ$ to -75° during their growing season [13], while the normal vector of fixed-tilt monofacial solar panels has a constant ideal elevation angle $\theta \geq +45^\circ$, depending mainly on latitude [1].

Can these differences explain the large difference between the energy-maximizing (i.e. ideal) eastern azimuth direction of mature sunflower inflorescences and the southern azimuth of fixed-tilt monofacial solar panels?

Discussion:

Interestingly, there is some similarity between the energy-maximizing ideal azimuth direction of fixed-tilt monofacial solar panels and that of non-heliotropic (i.e. non-sun-tracking) mature sunflower inflorescences: in both cases the ideal azimuth turns eastward, if afternoons are cloudier than mornings. This similarity exists despite the different absorption spectra, elevation angles, and activity period of sunflower inflorescences and solar panels. Mature sunflower inflorescences absorb light only in the summer months, during the flowering period between anthesis and senescence, and their ideal azimuth turns eastward. Depending on the regional cloud conditions, mature sunflower inflorescences facing east receive 54-77 % more energy than those facing south, if afternoons are generally cloudier than mornings, as is typical in the cultivation regions of sunflowers. This excess light energy is an obvious ecological advantage of east facing compared to south facing. Note that the analogy between sunflowers and fixed-tilt monofacial solar panels is valid only for the non-heliotropic mature sunflower inflorescences, rather than for the whole plant including the heliotropic (sun-tracking) leaves and immature bud.

References:

11. Shell GSG, Lang ARG, Sale PJM. 1974 Quantitative measures of leaf orientation and heliotropic response in sunflower, bean, pepper and cucumber. *Agricultural Meteorology* **13**, 25-37.
12. Rey H, Dautat J, Chenu K, Barczi JF, Dosio GAA, Lecoer J. 2008 Using a 3-D virtual sunflower to simulate light capture at organ, plant and plot levels: Contribution of organ interception, impact of heliotropism and analysis of genotypic differences. *Annals of Botany* **101**, 1139-1151.
14. Blackman BK, Scascitelli M, Kane NC, Luton HH, Rasmussen DA, Bye RA, Lentz DL, Rieseberg LH. 2011 Sunflower domestication alleles support single domestication center in eastern North America. *Proceedings of the National Academy of Sciences of the USA* **108**, 14360-14365. (doi: 10.1073/pnas.1104853108)

To the end of the revised Introduction section we added the following sentence inspired by the Editor:

Here we demonstrate that there may be some practical advantage to varying the orientation of photovoltaic solar panels from direct south/north, if there is systematic morning/afternoon asymmetry in cloudiness.

In the Discussion section, the paragraph dealing with the temperature effect was revised as follows:

Second, the warming up of solar panels is known to degrade electric output, because conversion efficiency drops with temperature [28, 29, 30, 31]. Vaillon et al. [29] listed three options to mitigate thermal effects in photovoltaic electric energy conversion. The first is to maximize cooling, the second is to minimize the thermal load in the panel, and the third is to minimize the thermal sensitivity of the electrical power output. In our current calculations, the temperature dependence of a solar cell's power generation efficiency is neglected. This conversion efficiency decreases/increases by 0.2-0.5 % for every 1 °C increase/decrease in temperature above/below the 25 °C reference temperature used in standard test conditions [32]. The operational cell temperature is higher than the ambient air temperature during daytime and primarily depends on the thermal properties of the cell material, the geometry and orientation of the panel, the type of the background surface (roof, wall, or open field), the solar insolation, and the amount of ventilation, which in turn depends on wind speed. Therefore, the actual temperature variation is highly location and installation specific. Nevertheless, both weather data-based thermal modelling studies and long-duration outdoor tests indicate that the diurnal cycle of cell temperature is usually skewed towards the afternoon, even in cloudy conditions: that is, the cell temperature is generally higher in the afternoon than in the morning [33, 34, 35]. The resulting (opposite) asymmetry in conversion efficiency, similar to the asymmetry in cloudiness and aerosol load, favours the morning, that is, the eastern hemisphere. Although a thermal loss of around 0.1-0.5%/K does not seem to be dramatic, it nevertheless needs to be investigated whether or not the 'optimal' orientation of solar panels – narrowly defined in the current study as the azimuth that maximizes the available solar energy – actually has a net positive effect on electric output. Such an empirical study is deferred to future research.

28. Dupre O, Vaillon R, Green MA. 2017 *Thermal Behavior of Photovoltaic Devices. Physics and Engineering*. Springer: Heidelberg, Berlin, New York
29. Vaillon R, Dupre O, Cal RB, Calaf M. 2018 Pathways for mitigating thermal losses in solar photovoltaics. *Scientific Reports* **8**, 13163. (doi: 10.1038/s41598-018-31257-0)
30. Vaillon R, Parola S, Lamnatou C, Chemisana D. 2020 Solar cells operating under thermal stress. *Cell Reports Physical Science* **1**, 100267. (doi: 10.1016/j.xcrp.2020.100267)
31. Kurtz S, Whitfield K, TamizhMani G, Koehl M, Miller D, Joyce J, Wohlgemuth J, Bosco N, Kempe M, Zgonena T. 2011 Evaluation of high-temperature exposure of photovoltaic modules. *Progress in Photovoltaics* **19**, 954-965. (doi: 10.1002/pip.1103)

Furthermore, as suggested by the Editor, the last sentence of the revised Discussion section sounds:

We emphasize that the implications of any final conclusion on the ideal azimuth angle of PV panels is the subject of further scrutiny of the variation of PV efficiency with respect to temperature and dust/sand/debris cover of solar panels, the impact of which was not investigated in the present study.

Subject Editor, Peter Haynes wrote: *I recommend the comment that seems to suggest that the fact that solar panel farms are (according to the authors) largely built in desert regions implies that cloudiness MUST outweigh temperature in important in this respect -- the logic seems flawed and, again, the point does not seem important to conclusions of your paper.*

Answer: As suggested by the Editor, in our revised Discussion section we deleted the following paragraph:

Because industrial solar panel farms are predominantly installed in regions with minimal cloudiness (e.g. deserts), the effect of higher panel temperatures must have a smaller impact on the energy-maximizing

ideal azimuth angle than the frequency of clouds: The panel's energy gain G due to the more intense direct solar radiation (in the absence of clouds) is greater than its energy loss L due to the smaller power conversion efficiency at higher temperatures ($G > L$). Otherwise (if $G < L$), industrial-scale solar panels would mainly be installed in regions with frequent clouds, which is not the case.

Subject Editor, Peter Haynes wrote: *I hope that the above recommendations provide a simple and reasonable approach to rapid publication of your paper in a form in which the major concrete conclusions remain as you intended. (If you made these changes then I would not see any reason to send the paper to the referee or referees once again.)*

Answer: In our 3rd revised manuscript we performed all of the above changes suggested by you.

Our manuscript was revised on the basis of the 3rd report of Reviewer 2. All changes suggested by **Referee 2 and the Editor** are marked with **orange**. We also wrote our detailed Point-by-Point Response to the comments of Referee 2.

I would like to submit our 3rd revised manuscript to *Royal Society Open Science*.

With best wishes: Gabor Horvath (corresponding author)

Prof. Gabor Horvath
Environmental Optics Laboratory,
Department of Biological Physics,
Eotvos University,
H-1117 Budapest, Pazmany setany 1, Hungary
e-mail: gh@arago.elte.hu
<http://arago.elte.hu>

Point-by-Point Response to the 3rd Review of Referee 2

We thank the 3rd review of Referee 2. Our manuscript was revised on the basis of this review. All changes suggested by Referee 2 and the Editor are marked with orange. Below is our detailed Point-by-Point Response to the comments of Referee 2.

Referee 2 wrote: *A) Sunflower vs solar cells: I respectfully disagree. The comparison between plants and solar panels is more distracting than helpful for the following reasons. Also, citations are often used to backup claims made in a manuscript. Any manuscript, however, should stand for itself and be accessible for most interested readers, without studying any (outdated) citations first.*

1. Fixed solar cells must maximise the insolation over a 365 day period, whereas sunflowers follow or see the sun over a much shorter time frame. The 'ideal' azimuth angle for sunflowers is therefore defined over a very different time span than the ideal azimuth for solar panels. Hence, they're not directly comparable, because they've different meanings.

2. As the flowers develop, they lose their flexibility of movement (for optimising their hourly insolation), such that the stems of mature sunflowers become stiffer and stationary (which might be optimised for seasonal insolation).

3. Many new varieties of sunflowers are bred so that the flower heads droop groundward as the plants mature. So birds cannot remove seeds as easily while the potential for diseases is reduced (caused by water collecting in the flower head). If such a downward tilted 'head contributes more than 25% of the whole-plant light absorption at maturity', a mature sunflower is probably not prioritising to maximise its insolation anymore.

4. Solar panels produce electrical energy; plants produce chemical energy. To which energy does the ideal azimuth refer? It is a little of comparing apples to oranges,

<https://www.scientificamerican.com/article/plants-versus-photovoltaics-at-capturing-sunlight/>

Sunflowers produce energy not specifically from sunlight but through a chemical breakdown of bonds that hold molecules together. They also are not 100 percent reliant on sunshine for energy production. In fact, the efficiency of photosynthesis is less than 3% (and indeed plants are not black). They can use soil nutrients in conjunction with sunlight and water to make energy. That means they do not need as much sunlight since their recipe for energy is broader than that of a solar panel. Hence, their energy needs are also limited. It is in this respect that I still have great difficulties to understand the comparison between the 'energy-maximising' azimuth direction of sunflowers and solar panels. I recommend to drop the comparison between apples and oranges entirely.

Answer: As suggested by Referee 2 and the Editor, in our 3rd revised manuscript we deleted the following **comparison between sunflowers and solar panels**:

Introduction:

On the other hand, after anthesis the non-heliotropic (i.e. tracking no longer the Sun) mature inflorescences of sunflowers (*Helianthus annuus*) face east. Only young sunflowers do track the sun during the day from east to west at their bud stage. The leaves of sunflowers continue sun-tracking to a dampened extent [11] after the azimuth direction of the mature inflorescences becomes fixed toward east, because the whole plant needs henceforward light energy for its physiological processes and the development of the growing head. After anthesis, the rapidly growing mature inflorescences need to absorb maximal light energy for the development of their blossoms and seeds. The head contributes more than 25% of the whole-plant light absorption at maturity [12]. Using an atmospheric radiation model with measured cloudiness and plant-physiological input data, Horváth *et al.* [13] showed that mature sunflower inflorescences absorb maximal light energy, if they face geographical east and the afternoons

are usually cloudier than the mornings in summer, as is the case in the area from which domesticated sunflowers originate [14]. Thus, ‘ideal azimuth angle’ of mature sunflower inflorescences means that they absorb maximal light energy if facing this azimuth direction.

There is some similarity between constantly east-facing mature sunflower inflorescences and fixed-tilt monofacial solar panels: both absorb as much light energy as possible with a fixed azimuth direction. However, there are three main differences between them:

- The absorption spectra of sunflower inflorescences and solar panels are different.
- Solar panels absorb sun/skylight throughout the year, while mature sunflower inflorescences absorb light only in their 2-3-month summer growing season between anthesis and senescence.
- The average elevation angle θ of the normal vector of mature sunflower inflorescences relative to the horizontal gradually decreases from about $+10^\circ$ to -75° during their growing season [13], while the normal vector of fixed-tilt monofacial solar panels has a constant ideal elevation angle $\theta \geq +45^\circ$, depending mainly on latitude [1].

Can these differences explain the large difference between the energy-maximizing (i.e. ideal) eastern azimuth direction of mature sunflower inflorescences and the southern azimuth of fixed-tilt monofacial solar panels?

Discussion:

Interestingly, there is some similarity between the energy-maximizing ideal azimuth direction of fixed-tilt monofacial solar panels and that of non-heliotropic (i.e. non-sun-tracking) mature sunflower inflorescences: in both cases the ideal azimuth turns eastward, if afternoons are cloudier than mornings. This similarity exists despite the different absorption spectra, elevation angles, and activity period of sunflower inflorescences and solar panels. Mature sunflower inflorescences absorb light only in the summer months, during the flowering period between anthesis and senescence, and their ideal azimuth turns eastward. Depending on the regional cloud conditions, mature sunflower inflorescences facing east receive 54-77 % more energy than those facing south, if afternoons are generally cloudier than mornings, as is typical in the cultivation regions of sunflowers. This excess light energy is an obvious ecological advantage of east facing compared to south facing. Note that the analogy between sunflowers and fixed-tilt monofacial solar panels is valid only for the non-heliotropic mature sunflower inflorescences, rather than for the whole plant including the heliotropic (sun-tracking) leaves and immature bud.

References:

11. Shell GSG, Lang ARG, Sale PJM. 1974 Quantitative measures of leaf orientation and heliotropic response in sunflower, bean, pepper and cucumber. *Agricultural Meteorology* **13**, 25-37.
12. Rey H, Dautat J, Chenu K, Barczi JF, Dosio GAA, Lecoer J. 2008 Using a 3-D virtual sunflower to simulate light capture at organ, plant and plot levels: Contribution of organ interception, impact of heliotropism and analysis of genotypic differences. *Annals of Botany* **101**, 1139-1151.
14. Blackman BK, Scascitelli M, Kane NC, Luton HH, Rasmussen DA, Bye RA, Lentz DL, Rieseberg LH. 2011 Sunflower domestication alleles support single domestication center in eastern North America. *Proceedings of the National Academy of Sciences of the USA* **108**, 14360-14365. (doi: 10.1073/pnas.1104853108)

Referee 2 wrote: *B) Temperature vs cloud coverage: 'The effect of higher panel temperatures must have a smaller impact on the energy-maximizing ideal azimuth angle than the frequency of clouds, because industrial solar panel farms are predominantly installed in regions with minimal cloudiness.'*

Following this train of thought, the quality of air must have a smaller impact on life expectancy than crossing the street by red, driving too fast or after a glass of wine, because many (adult) people are often ignoring the traffic lights, speed limits or BAC levels.

But even if the authors can provide data in support of their claim, e.g. cloudiness-index (its annual average) vs latitude, it would not be sufficient to explain the causation of a correlation. In deserts, as far as I understood, apart from the high temperatures, dust/sand accumulation on the solar panels is one of the greatest concerns. If so, the frequency of cleaning the panels will likely be more important than their azimuth angle.

Finally, colleagues at Tampere University looked into how cloud transitions affect the performance of real-world PV systems. For example, the irradiance incident on PV generators can considerably EXCEED the expected clear sky irradiance. Due to this phenomenon, called cloud enhancement (CE), the maximum power of the PV generator can exceed the rated power of the inverter connecting the generator to the grid, <https://doi.org/10.1063/5.0007550>. But more importantly, often the impact of cloud transitions on a PV system can simply be ignored (especially for large-scale systems), <http://dx.doi.org/10.1049/iet-rpg.2019.0085>, <https://doi.org/10.1016/j.renene.2020.01.119>.

Yes, cloud coverage will reduce the annual insolation (and thus PV solar yield) in overall, but 'cloudy' countries are often also characterised by stronger winds and cooler temperatures, e.g. Ireland, while rain clouds actually help to keep solar panels clean and mostly free from debris. There is a reason for why solar farms are installed in the UK, too. Cloudy Norway is even further north, yet it is quite possible to produce solar energy there: ...s, a small town south of Oslo, receives 1000 kilowatt-hours (kWh) per square meter annually. This is comparable to many parts of Germany, where solar power has boomed over the last 10 years. Last but not least, if cloud coverage would be a greater impediment to solar PV installations than temperature, floating solar farms should have a darker and by far less brighter future.

In summary, the argument for why cloudiness (and the frequency of cloud transitions) should have a greater impact than temperature [on the optimum/ideal azimuth direction of solar panels] is here too weakly presented by the authors, since it requires a thoroughly elaborated discussion with numbers (data).

Answer: Inspired by the above comments of Referee 2, to the end of the Discussion section of our 3rd revised manuscript we added the following paragraphs:

Cloud cover reduces the annual insolation, and thus PV solar yield in general. However, clouds influence the efficiency of solar panels in other ways, too. In deserts, apart from the high temperatures (decreasing the efficiency of PV panels), dust and sand accumulation on the panels (decreasing the light intensity available for panels) is also of great concern. Depending on the frequency of windy conditions, the dust/sand-covered panels should periodically be cleaned, which is a time-consuming and expensive activity. On the other hand, countries with cloudy climates (e.g. Ireland, England, Scandinavia) usually experience stronger winds, cooler air temperatures, and more frequent rain (which cleans the solar panels); these factors increase PV electric output all else being equal.

Cloud transitions also affect the performance of PV systems. The irradiance incident on PV generators can considerably exceed the expected clear sky irradiance, which phenomenon is called cloud enhancement (CE) [36]. Due to CE, the maximum power of the PV generator can exceed the rated power of the inverter connecting the generator to the grid. It was shown that the effect of CE is small on the aggregated energy, because CE events that most strongly impact PV system operations are very rare[36].

The fast irradiance transitions caused by clouds are partial shading events that cause fast power fluctuations leading even to stability and quality problems in power networks [37]. Fast non-homogeneous irradiance transitions also cause mismatch losses in PV generators and the occurrence of multiple maximum power points (MPPs), which appear in a wide voltage range of the PV generator. It was demonstrated that the energy losses due to operation at a local MPP instead of the global one during partial shading events by clouds have only a minor effect on the total energy production of PV arrays, especially for large-scale systems [38].

In an improved model of our computational approach, the above effects can also be taken into consideration to determine the performance-maximizing (rather than the insolation-maximizing) locally ideal azimuth angle of solar panels.

36. Lappalainen K, Kleissl J. 2020 Analysis of the cloud enhancement phenomenon and its effects on photovoltaic generators based on cloud speed sensor measurements. *Journal of Renewable and Sustainable Energy* **12**, 043502. (doi: 10.1063/5.0007550)
37. Lappalainen K, Valkealahti S. 2019 Fluctuation of PV array global maximum power point voltage during irradiance transitions caused by clouds. *IET Renewable Power Generation* **13**, 2864-2870. (doi: 10.1049/iet-rpg.2019.0085)
38. Lappalainen K., Valkealahti S. 2020 Number of maximum power points in photovoltaic arrays during partial shading events by clouds. *Renewable Energy* **152**, 812-822. (doi: 10.1016/j.renene.2020.01.119)